# The RL Perceptron: Generalisation Dynamics of Policy Learning in High Dimensions

## Abstract

Reinforcement learning (RL) algorithms have proven transformative in a range of domains. To tackle real-world domains, these systems often use neural networks to learn policies directly from pixels or other high-dimensional sensory input. By contrast, much theory of RL has focused on discrete state spaces or worst-case analysis, and fundamental questions remain about the dynamics of policy learning in high-dimensional settings. Here, we propose a solvable high-dimensional model of RL that can capture a variety of learning protocols, and derive its typical dynamics as a set of closed-form ordinary differential equations (ODEs). We derive optimal schedules for the learning rates and task difficulty—analogous to annealing schemes and curricula during training in RL—and show that the model exhibits rich behaviour, including delayed learning under sparse rewards; a variety of learning regimes depending on reward baselines; and a speed-accuracy trade-off driven by reward stringency. Experiments on a variant of the Procgen game "Bossfight" also show such a speed-accuracy trade-off in practice. Together, these results take a step towards closing the gap between theory and practice in high-dimensional RL.

Recent years have seen rapid progress in Reinforcement Learning (RL): algorithmic and engineering breakthroughs led to super-human performance in a variety of domains, for example complex games like Go [Silver et al., 2016, Mnih et al., 2015]. Despite these practical successes, our theoretical understanding of RL for high-dimensional problems requiring non-linear function approximation is still limited. While comprehensive theoretical results exist for tabular RL, where the state and action spaces are discrete and small enough for value functions to be represented directly, the curse of dimensionality limits these methods to low-dimensional problems. The lack of a clear notion of similarity between discrete states further means that tabular methods do not address the core question of generalisation: how are values and policies extended to unseen states and across seen states [Kirk et al., 2023]? As a consequence, much of this theoretical work is far from the current practice of RL, which increasingly relies on deep neural networks to approximate and generalise value functions, policies and other building blocks of RL. Moreover, while RL theory has often addressed "worst-case" performance and convergence behaviour, the *typical* behaviour has received comparatively little attention (cf. further related work below). Meanwhile, a growing sub-field of deep learning theory has employed tools from statistical mechanics to analyse various supervised learning paradigms in the average-case, see Seung et al. [1992], Engel and Van den Broeck [2001], Carleo et al. [2019], Bahri et al. [2020], Gabrié et al. [2023] for classical and recent reviews. While this approach has recently been extended to curriculum learning [Saglietti et al., 2022], continual learning [Asanuma et al., 2021, Lee et al., 2021, 2022], few-shot learning [Sorscher et al., 2022] and transfer learning [Lampinen and Ganguli, 2018, Dhifallah and Lu, 2021, Gerace et al., 2022], RL has not been analysed yet using statistical mechanics—a gap we address here by studying the high-dimensional generalisation dynamics of a simple neural network trained on a reinforcement learning task.

Submitted to 37th Conference on Neural Information Processing Systems (NeurIPS 2023). Do not distribute.

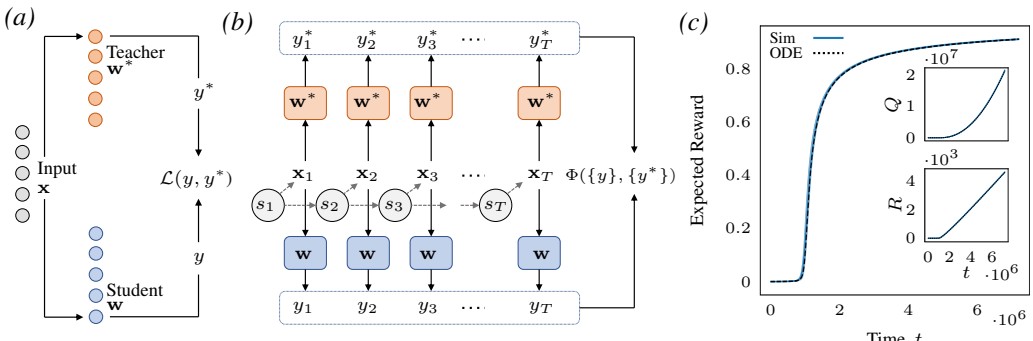

Figure 1: **The RL-Perceptron is a model for policy learning in high dimensions.** *(a)* In the classic teacher-student model for supervised learning, a neural network called the student is trained on inputs $x$ whose label $y^*$ is given by another neural network, called the teacher. *(b)* In the RL setting the student moves through states $s_t$ making a series of $T$ choices given in response to inputs $x_t$. The RL-perceptron is an extension of the teacher-student model as we assume there is a 'right' choice $y_t$ on each timestep given by a teacher network. The student receives a reward after $T$ decisions according to a criterion $\Phi$ that depends on the choices made and the corresponding correct choices. *(c)* Example learning dynamics in the RL-perceptron for a problem with $T = 12$ choices where the reward is given only if all the decisions are correct. The plot shows the expected reward of a student trained in the RL perceptron setting in simulations (solid) and for our theoretical results (dashed) obtained from solving the dynamical equations eqs. (5) and (6). Finite size simulations and theory show good agreement. We reduce the stochastic evolution of the high dimensional student to the study of deterministic evolution of two scalar quantities $R$ and $Q$ (more details in Sec. 2.1), their evolution are shown in the inset. *Parameters: $D = 900$, $\eta_1 = 1$, $\eta_2 = 0$, $T = 12$.*

**The RL perceptron:** In the classic teacher-student model of supervised learning [Gardner and Derrida, 1989, Seung et al., 1992], a neural network called the student is trained on inputs $\mathbf{x}$ whose labels $y^*$ are given by another neural network called the teacher (see fig. 1a). The goal of the student is to learn the function represented by the teacher from samples $(\mathbf{x}, y^*)$. In RL, agents face a sequential decision-making task in which a sequence of correct intermediate choices is required to successfully complete an episode. We translate this process into the RL perceptron, a solvable model for a high-dimensional, sequential policy learning task shown in fig. 1b. The student with weights $\mathbf{w}$ takes a sequence of $T$ choices over an episode. The correct choices are governed by the same teacher network $\mathbf{w}^*$, i.e. the same underlying rule throughout every time-step of every episode. Crucially, unlike in the supervised learning setting, the student does not observe the correct choice for each input; instead, it receives a reward which depends on whether earlier decisions are correct. For instance, the student could receive a reward only if all $T$ choices are correct, and no reward otherwise—a learning signal that is considerably less informative than in supervised learning. In addition to introducing the RL perceptron, our **main contributions** are as follows:

- We derive an asymptotically exact set of Ordinary Differential Equations (ODEs) that describe the typical learning dynamics of policy gradient RL agents by building on classic work by Saad and Solla [1995], Biehl and Schwarze [1995], see section 2.1.

- We use these ODEs to characterize learning behaviour in a diverse range of scenarios:
    - We explore several sparse delayed reward schemes and investigate the impact of negative rewards (section 2.2)
    - We derive optimal learning rate schedules and episode length curricula, and recover annealing strategies typically used in practice (section 2.3)
    - At fixed learning rates, we identify ranges of learning rates for which learning is 'easy,' and 'hybrid-hard'—possibly causing a critical slowing down in the dynamics (section 2.4)
    - We identify a speed-accuracy trade-off driven by reward stringency (section 2.5)

- Finally we demonstrate that a similar speed-accuracy trade-off exists in simulations of high-dimensional policy learning from pixels using the procgen environment "Bossfight" [Cobbe et al., 2019], see section 3.

## Further related work

**Sample complexity in RL.** An important line of work in the theory of RL focuses on the sample complexity and other learnability measures for specific classes of models such as tabular RL [Azar et al., 2017, Zhang et al., 2020b], state aggregation [Dong et al., 2019], various forms of MDPs [Jin et al., 2020, Yang and Wang, 2019, Modi et al., 2020, Ayoub et al., 2020, Du et al., 2019a, Zhang et al., 2022], reactive POMDPs [Krishnamurthy et al., 2016], and FLAMBE [Agarwal et al., 2020]. Here, we are instead concerned with the learning dynamics: how do reward rates, episode length, etc. influence the speed of learning and the final performance of the model.

**Statistical learning theory for RL** aims at finding complexity measures analogous to the Rademacher complexity or VC dimension from statistical learning theory for supervised learning Bartlett and Mendelson [2002], Vapnik and Chervonenkis [2015]. Proposals include the Bellman Rank Jiang et al. [2017], or the Eluder dimension [Russo and Van Roy, 2013] and its generalisations [Jin et al., 2021]. This approach focuses on worst-case analysis, which typically differs significantly from practice (at least in supervised learning [Zhang et al., 2021]). Furthermore, complexity measures for RL are generally more suitable for value-based methods; policy gradient methods have received less attention despite their prevalence in practice Bhandari and Russo [2019], Agarwal et al. [2021]. We focus instead on average-case dynamics of policy-gradient methods.

**Dynamics of learning.** A series of recent papers considered the dynamics of temporal-difference learning and policy gradient in the limit of wide two-layer neural networks Cai et al. [2019], Zhang et al. [2020a], Agazzi and Lu [2021, 2022]. These works focus on one of two "wide" limits: either the neural tangent kernel [Jacot et al., 2018, Du et al., 2019b] or "lazy" regime [Chizat et al., 2019], where the network behaves like an effective kernel machine and does not learn data-dependent features, which is key for efficient generalisation in high-dimensions. In our setting, the success of the student crucially relies on learning the weight vector of the teacher, which is hard for lazy methods [Ghorbani et al., 2019, 2020, Chizat and Bach, 2020, Refinetti et al., 2021]. The other "wide" regime is the mean-field limit of interacting particles, akin to Mei et al. [2018], Chizat and Bach [2018], Rotskoff and Vanden-Eijnden [2018], where learning dynamics are captured by a non-linear partial differential equation. While this elegant description allows them to establish global convergence properties, it is hard to solve in practice. The ODE description we derive here instead will allow us to describe a series of effects in the following sections.

## 1 The RL Perceptron: setup and learning algorithm

We study the simplest possible student network, a perceptron with weight vector $\mathbf{w}$ that takes in high-dimensional inputs $\mathbf{x} \in \mathbb{R}^D$ and outputs $y(\mathbf{x}) = \mathrm{sgn}(\mathbf{w}^\intercal \mathbf{x})$. We interpret the outputs $y(\mathbf{x})$ as decisions, for example whether to go left or right in an environment. Because the student makes choices in response to high-dimensional inputs, it is analogous to a policy network. To train the network, we therefore consider a policy gradient learning update analogous to the REINFORCE algorithm [Sutton et al., 2000] that is adapted to the perceptron. At every timestep $t$ during the $\mu$th episode of length $T$, the agent occupies some state $s_t$ in the environment, receives an observation $\mathbf{x}_t^\mu$ conditioned on $s_t$, and takes an action $y_t^\mu = \mathrm{sgn}(\mathbf{w}^{\mu\intercal}\mathbf{x}_t^\mu)$, with $t = 1, \ldots, T$. The correct choice for each input is given by a fixed perceptron teacher with weights $\mathbf{w}^*$. The crucial point is that the student does not have access to all the correct choices; it only receives a reward at the end of the episode *if* it completes the episode successfully, for example by making the correct decision at all times. If it does not succeed, it *may* receive a penalty; we will see in section 2.4 that receiving penalties is not always beneficial. In our setup, this translates into a weight update at the end of the $\mu^{\text{th}}$ episode that is given by

$$\mathbf{w}^{\mu+1} = \mathbf{w}^\mu + \frac{\eta_1}{\sqrt{D}}\left(\frac{1}{T}\sum_{t=1}^{T} y_t \mathbf{x}_t \mathbb{I}(\Phi)\right)^\mu - \frac{\eta_2}{\sqrt{D}}\left(\frac{1}{T}\sum_{t=1}^{T} y_t \mathbf{x}_t (1 - \mathbb{I}(\Phi))\right)^\mu, \tag{1}$$

where $\mathbb{I}$ is an indicator function and $\Phi$ is the criterion that determines whether the episode was completed successfully—for instance, $\mathbb{I}(\Phi) = \prod_t^T \theta(y_t y_t^*)$ (where $\theta$ is the step function) if the student has to get every decision right in order to receive a reward. The update is general in the sense that the term proportional to the learning rate $\eta_1 > 0$ prescribes the reward update for the fulfillment of the condition, while the term proportional to $\eta_2 \geq 0$ gives us the possibility to add a a penalty or negative

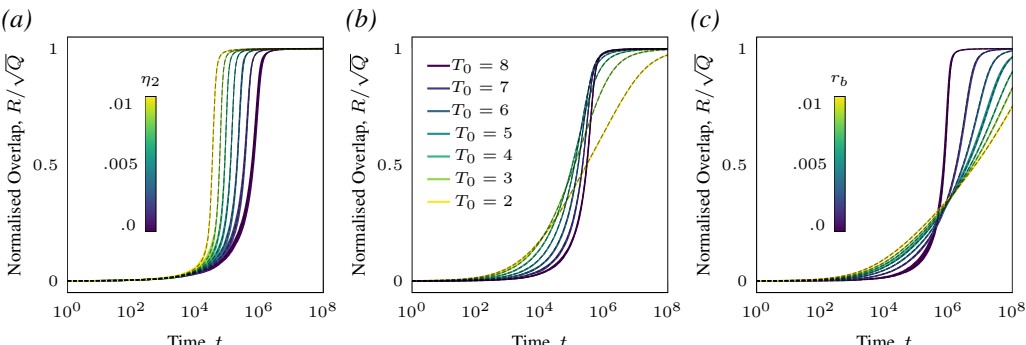

Figure 2: **ODEs accurately describe diverse learning protocols.** Evolution of the normalised student-teacher overlap $\rho$ for the numerical solution of the ODEs (dashed) and simulation (coloured) in three reward protocols. All students receive a reward of $\eta_1$ for getting all decisions in an episode correct, and additionally: *(a)* A penalty $\eta_2$ (i.e. negative reward) is received if the agent does not survive until the end of an episode. *(b)* An additional reward of 0.2 is received if the agent survives beyond $T_0$ timesteps. *(c)* An additional reward $r_b$ is received for every correct decision made in an episode. *Parameters:* $D = 900$, $T = 12$, $\eta_1 = 1$.

reward should the student not succeed. Note that in the case of $T = 1$, $\eta_2 = 0$, and $\mathbb{I}(\Phi) = \theta(yy^*)$, the learning rule updates the weight only if the student is correct on a given sample. It can thus be seen as the "opposite" of the famous perceptron learning rule of supervised learning [Rosenblatt, 1962], where weights are only updated if the student is wrong. For a more in-detail discussion of the relation between the weight update in eq. (1) and the REINFORCE algorithm, see appendix A.

## 2 Theoretical Results

### 2.1 A set of dynamical equations captures the learning dynamics of an RL perceptron exactly

The goal of the student during training is to emulate the teacher as closely as possible; or in other words, have a small number of disagreements with the teacher $y(\mathbf{x}) \neq y^*(\mathbf{x})$. The generalisation error is given by the average number of disagreements

$$\epsilon_g \equiv \langle y(\mathbf{x})y^*(\mathbf{x}) \rangle = \left\langle \text{sgn}\left(\mathbf{w}^* \cdot \mathbf{x}/\sqrt{D}\right) \text{sgn}\left(\mathbf{w} \cdot \mathbf{x}/\sqrt{D}\right)\right\rangle = \langle \text{sgn}(\nu)\text{sgn}(\lambda)\rangle \qquad (2)$$

where the average $\langle \cdot \rangle$ is taken over the inputs $\mathbf{x}$, and we have introduced the scalar pre-activations for the student and the teacher, $\lambda \equiv \mathbf{w} \cdot \mathbf{x}/\sqrt{D}$ and $\nu \equiv \mathbf{w}^* \cdot \mathbf{x}/\sqrt{D}$, respectively. We can therefore transform the high-dimensional average over the inputs $\mathbf{x}$ into a low-dimensional average over the pre-activations $(\lambda, \nu)$. The average in eq. (2) can be carried out by noting that the tuple $(\lambda, \nu)$ follow a jointly Gaussian distribution with means $\langle \lambda \rangle = \langle \nu \rangle = 0$ and covariances

$$Q \equiv \langle \lambda^2 \rangle = \frac{\mathbf{w} \cdot \mathbf{w}}{D}, \quad R \equiv \langle \lambda\nu \rangle = \frac{\mathbf{w} \cdot \mathbf{w}^*}{D} \quad \text{and} \quad S \equiv \langle \nu^2 \rangle = \frac{\mathbf{w}^* \cdot \mathbf{w}^*}{D}. \qquad (3)$$

These covariances, or overlaps as they are sometimes called in the literature, have a simple interpretation. The overlap $S$ is simply the length of the weight vector of the teacher; in the high-dimensional limit $D \to \infty$, $S \to 1$. Likewise, the overlap $Q$ gives the length of the student weight vector; however, this is a quantity that will vary during training. For example, when starting from small initial weights, $Q$ will be small, and grow throughout training. Lastly, the "alignment" $R$ quantifies the correlation between the student and the teacher weight vector. At the beginning of training, $R \approx 0$, as both the teacher and the initial condition of the student are drawn at random. As the student starts learning, the overlap $R$ increases. Evaluating the Gaussian average in eq. (2) shows that the generalisation error is then a function of the normalised overlap $\rho = R/\sqrt{Q}$, and given by

$$\epsilon_g = \frac{1}{\pi} \arccos\left(\frac{R}{\sqrt{Q}}\right) \qquad (4)$$

The crucial point here is that we have reduced the description of the high-dimensional learning problem from the $D$ parameters of the student weight $\mathbf{w}$ to two time-evolving quantities, $Q$ and $R$. We now discuss how to analyse their dynamics.

**The dynamics of order parameters.** At any given point during training, the value of the order parameters determines the test error via eq. (4). But how do the order parameters evolve during training with the update rule eq. (1)? We followed the approach of Kinzel and Ruján [1990], Saad and Solla [1995], Biehl and Schwarze [1995] to derive a set of dynamical equations that describe the dynamics of the student in the high-dimensional limit where the input dimension goes to infinity. We give explicit dynamics for different reward conditions $\Phi$, namely requiring all decisions correct in an episode of length $T$; requiring $n$ or more decisions correct in an episode of length $T$; and receiving reward for each correct response. Due to the length of these expressions, we report the generic expression of the updates in the supplementary material in appendix B. Below, we state a version of the equations for the specific reward condition where the agent must survive until the end of an episode to receive a reward, $\mathbb{I}(\Phi) = \prod_t^T \theta(y_t y_t^*)$. The ODEs for the order parameters then read

$$\frac{dR}{d\alpha} = \frac{\eta_1 + \eta_2}{\sqrt{2\pi}} \left(1 + \frac{R}{\sqrt{Q}}\right) P^{T-1} - \eta_2 R \sqrt{\frac{2}{\pi Q}} \tag{5}$$

$$\frac{dQ}{d\alpha} = (\eta_1 + \eta_2)\sqrt{\frac{2Q}{\pi}} \left(1 + \frac{R}{\sqrt{Q}}\right) P^{T-1} - 2\eta_2 \sqrt{\frac{2Q}{\pi}} + \frac{(\eta_1^2 - \eta_2^2)}{T} P^T + \frac{\eta_2^2}{T}, \tag{6}$$

where $\alpha \equiv \mu/D$ serves as a continuous time variable in the limit $D \to \infty$ (not to be confused with $t$ which counts episode steps), and $P = \left(1 - \cos^{-1}(R/\sqrt{Q})/\pi\right)$ is the probability of a single correct decision. While our derivation of the equations follow heuristics from statistical physics, we anticipate that their asymptotic correctness in the limit $D \to \infty$ can be established rigorously using the techniques of Goldt et al. [2019], Veiga et al. [2022], Arnaboldi et al. [2023]. We illustrate the accuracy of these equations already in finite dimensions ($D = 900$) in fig. 1c, where we show the expected reward, as well as the overlaps $R$ and $Q$, of a student as measured during a simulation and from integration of the dynamical equations (solid and dotted lines, respectively).

The derivation of the dynamical equations that govern the learning dynamics of the RL perceptron are our first main result. Equipped with this tool, we now analyse several phenomena exhibited by the RL perceptron through a detailed study of these equations.

## 2.2 Learning protocols

The RL perceptron allows for the characterization of different RL protocols by adapting the reward condition $\Phi$. We considered the following three settings:

**Vanilla:** The dynamics in the 'standard' case without penalty, $\eta_2 = 0$, is shown in fig. 5a and fig. 5b. Rewards are sparsest in this protocol, and as a result we observe a characteristic initial plateau in expected reward followed by a rapid jump. The length of this plateau increases with $T$, consistent with the notion that sparse rewards make exploration hard and slow learning [Bellemare et al., 2016]. Plateaus during learning, which arise from saddle points in the loss landscape, have also been studied for (deep) neural networks in the supervised setting [Saad and Solla, 1995, Dauphin et al., 2014], but do not arise in the supervised perceptron. Hence the RL setting can qualitatively change the learning trajectory. The benefit of withholding penalties is that while slower, the perceptron reaches the highest level of expected reward in this case. This is a first example of a speed-accuracy trade-off that we will explore in more detail in section 2.5 and that we also found in our experiments with Bossfight in section 3.

**Penalty:** The initial plateau can be reduced by providing a penalty or negative reward ($\eta_2 > 0$) when the student fails in the task. This change provides weight updates much earlier in training and thus accelerates the escape from the plateau. The dynamics under this protocol are shown in fig. 2a. It is clear the penalty provides an initial speed-up in learning, as expected if the agent were to be unaligned and more likely to commit an error. However, a high penalty can create additional sub-optimal fixed points in the dynamics leading to a low asymptotic performance (more on this in section 2.4). In the simulations, finite size effects occasionally permit escape from the sub-optimal fixed point and jumps to the optimal one, leading to a high variance in the results.

**Subtask and breadcrumbs:** The model is also able to capture the dynamics of more complicated protocols: fig. 3b shows learning under the protocol where a smaller sub-reward is received if the agent survives beyond a shorter duration $T_0 < T$, i.e. some reward is still received even if the agent does not survive for the entire episode. Another learning protocol we can capture is that of 'graded-breadcrumbs', where the agent receives a small reward $r_b$ for every correct decision made

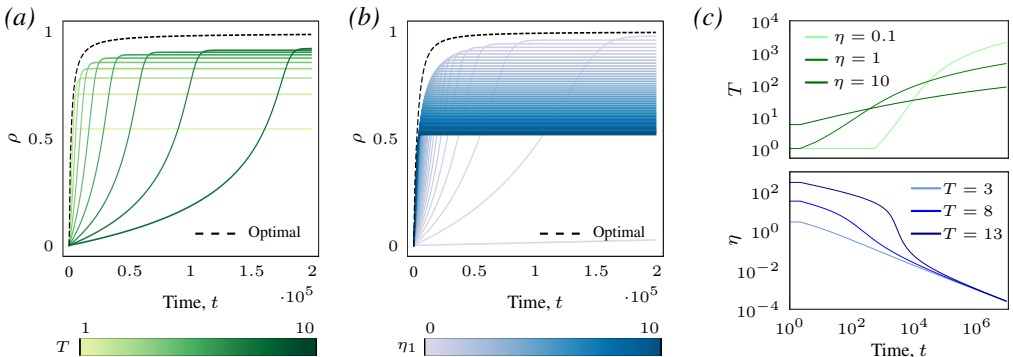

Figure 3: **Optimal schedules for episode length $T$ and learning rate $\eta$.** *(a)* Evolution of the normalised overlap under optimal episode length scheduling (dashed) and various constant episode lengths (green). *(b)* Evolution of the normalised overlap under optimal learning rate scheduling (dashed) and various constant learning rates (blue). *(c)* Evolution of optimal $T$ (green) and $\eta$ (blue) over learning. *Parameters: $D = 900$, $Q = 1$, $\eta_2 = 0$, (a) $\eta = 1$, (b) $T = 8$.*

in an episode, i.e. like the previous method some reward is still received even if the agent does not survive for the entire episode, these dynamics are captured in fig. 3c.

## 2.3 Optimal hyper-parameter schedules: make episodes longer and anneal your learning rate

Hyper-parameter schedules are crucial for successful training of RL agents. In our setup, the two most important hyper-parameters are the learning rates and the episode length. In the RL perceptron, we can derive optimal schedules for both hyper-parameters. For simplicity, here we report the results in the spherical case, where the length of the student vector is fixed at $\sqrt{D}$ (we discuss the unconstrained case in the appendix C), then $Q(\alpha) = 1$ at all times and we only need to track the teacher-student overlap $\rho = R/\sqrt{Q}$, which quantifies the generalisation performance of the agent. Keeping the choice $\mathbb{I}(\Phi) = \prod_{t=1}^{T} \theta(y_t y_t^*)$ and turning off the penalty term ($\eta_2 = 0$), we find that the teacher-student overlap is governed by the equation

$$\frac{d\rho}{d\alpha} = \frac{\eta}{\sqrt{2\pi Q}}(1 - \rho^2)\left(1 - \frac{1}{\pi}cos^{-1}(\rho)\right)^{T-1} - \frac{\eta^2}{2TQ}\rho\left(1 - \frac{1}{\pi}cos^{-1}(\rho)\right)^T \qquad (7)$$

The optimal schedules over episodes for $T$ and $\eta$ can then be found by maximising the change in overlap at each update, i.e. setting $\frac{\partial}{\partial T}\left(\frac{d\rho}{d\alpha}\right)$ and $\frac{\partial}{\partial \eta}\left(\frac{d\rho}{d\alpha}\right)$ to zero respectively. After some calculations, we find the optimal schedules to be

$$T_{\text{opt}} = \left\lfloor \frac{\sqrt{\pi}}{2}\frac{\eta\rho P}{(1-\rho^2)\sqrt{2Q}}\left[1 + \sqrt{1 - \frac{\sqrt{2Q}}{\eta\rho}\frac{4(1-\rho^2)}{\sqrt{\pi}P\ln(P)}}\right]\right\rfloor \quad \text{and} \quad \eta_{\text{opt}} = \sqrt{\frac{Q}{2\pi}}\frac{T(1-\rho^2)}{\rho P} \tag{8}$$

where $\lfloor \cdot \rfloor$ indicates the floor function.

Figure 3a shows the evolution of $\rho$ under the optimal episode length schedule (dashed) compared to other constant episode lengths (green). Similarly, fig. 3b shows the evolution of $\rho$ under the optimal learning rate schedule (dashed) compared to other constant learning rates (blue). The functional forms of $T_{\text{opt}}$ and $\eta_{\text{opt}}$ over time are shown in fig. 3c.

During learning the student seeks increasingly refined information to improve its expected reward. This simple observation explains the monotonic increase of the optimal episode length and the decrease in learning rates. Starting from the episode duration, we can observe that given the discrete nature of the decisions, information obtained from the rewards simply pushes the decision boundary towards a partition of the input space. This partition is determined by the episode length $T$ and correspond to a fraction $1/2^T$ of the entire input space. Therefore a positive reward conveys $T$ bits of information. At a fixed learning rate, when the student becomes proficient in the task it will not be able to improve further the decision boundary, and will fluctuate around the optimal solution unless longer episodes are provided.

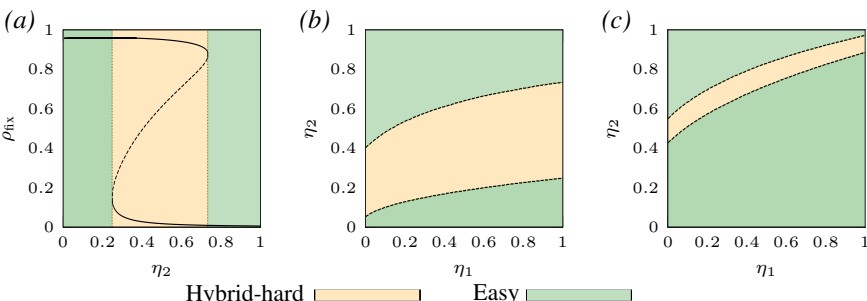

Figure 4: **Phase plots characterising learnability**. In the case where all decisions in an episode of length $T$ must be correct in order to receive a reward. *(a)* the fixed points of $\rho$ for $T = 13$ and $\eta_1 = 1$, the dashed portion of the line denotes where the fixed points are unstable. *(b)* Phase plot showing regions of hardness for $T = 13$. *(c)* Phase plot showing regions or hardness for $T = 8$. The green regions represent the *Easy* phase where with probability 1 the algorithm naturally converges to the optimal $\rho_{\text{fix}}$ from a random initialisation. The orange region indicates the *Hybrid-hard* phase, where with high probability the algorithm converges to the sub-optimal $\rho_{\text{fix}}$ from random initilisation. *Parameters: $D = 900$, $Q = 1$.*

Our analysis shows that a polynomial increase in the episode length gives the optimal performance in the RL perceptron, see fig. 3c (top); increasing $T$ in the RL perceptron is akin to increasing task difficulty, and the polynomial scheduling of $T_{\text{opt}}$ specifies a curriculum. Curricula of increasing task difficulty are commonly used in RL to give convergence speed-ups and learn problems that otherwise would be too difficult to learn *ab initio* Narvekar et al. [2020]. Analogously, the fluctuations can be reduced by annealing the learning rate and averaging over a larger number of samples. Akin to work in RL literature studying adaptive step-sizes [Dabney, 2014, Pirotta et al., 2013], we find that annealing the learning rate during training is beneficial for greater speed and generalisation performance. For the RL perceptron, a polynomial decay in the learning rate gives optimal performance as shown in fig. 3c (bottom), consistent with work in the parallel area of high-dimensional non-convex optimization problems [d'Ascoli et al., 2022], and stochastic approximation algorithms in RL [Dalal et al., 2017].

## 2.4 Phase Space

With a non-zero penalty ($\eta_2$), the generalisation performance of the agent can enter different regimes of learning. This is most clearly exemplified in the spherical case, where the number of fixed points of the ODE governing the dynamics of the overlap exist in distinct phases determined by the combination of reward and penalty. For the simplest case $\left( \mathbb{I}(\Phi) = \prod_t^T (y_t y_t^*) \right)$ these phases are shown in fig. 4. Figure 4a shows the fixed points achievable over a range of penalties for a fixed $\eta_1 = 1$ (obtained from a numerical solution of the ODE in $\rho$). There are two distinct regions: 1) *Easy*, where there is a unique fixed point and the algorithm naturally converges to this optimal $\rho_{\text{fix}}$ from a random initialisation, 2) a *Hybrid-hard* region (given the analogy with results from inference problems Ricci-Tersenghi et al. [2019]), where there are two stable (1 good and 1 bad) fixed points, and 1 unstable fixed point, and either stable point is achievable depending on the initialisation of the student (orange). The 'hybrid-hard' region separates two easy regions with very distinct performance levels. In this region the algorithm with high probability converges to $\rho_{\text{fix}}$ with the worse performance level. These two regions are visualised in $(\eta_1, \eta_2)$ space in fig. 4b for an episode length of $T = 13$. The topology of these regions are also governed by episode length, with a sufficiently small T reducing the the area of the 'hybrid-hard' phase to zero, meaning there is always 1 stable fixed point which may not necessarily give 'good' generalisation. Figure 4c shows the phase plot for $T = 8$, where the orange (hybrid-hard) has shrunk, this corresponds to the s-shaped curve in fig. 4a becoming flatter (closer to monotonic). Learning with $\eta_2$ This is not a peculiarity specific to the spherical case, indeed, we observe different regimes in the learning dynamics in the setting with unrestricted $Q$ which we report in appendix C.

These phases show that at a fixed $\eta_1$ increasing $\eta_2$ will eventually lead to a first order phase transition, and the speed benefits gained from a non-zero $\eta_2$ will be nullified due to the transition into the hybrid-hard phase. In fact, when taking $\eta_2$ close to the transition point, instead of speeding up learning there is the presence of a critical slowing down, which we report in appendix C.

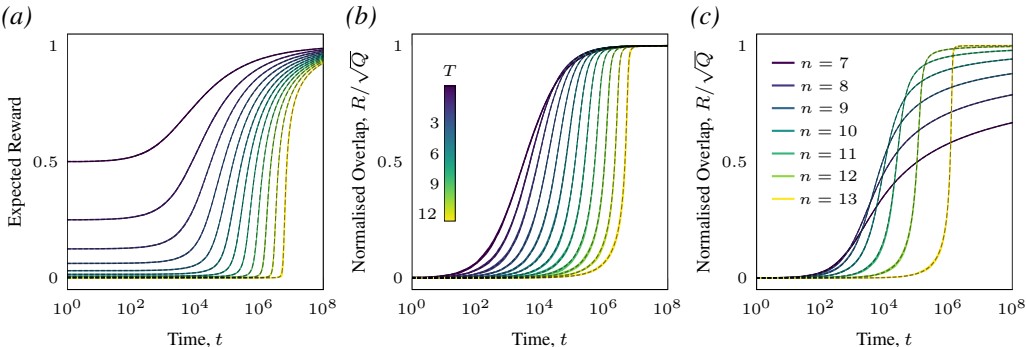

Figure 5: **Speed-accuracy tradeoff**. Evolution of *(a)* the expected reward and *(b)* corresponding normalised overlap for simulation (solid) and ODE solution (dashed) over a range of $T$ when all decisions in an episode of length $T$ are required correct, and $\eta_2 = 0$. *(c)* Evolution of the normalised overlap between student and teacher weights for simulation (solid) and ODE solution (dashed) for the case where $n$ or more decisions in an episode of length 13 are required correct for an update with $\eta_2 = 0$. More stringent reward conditions slow learning but can improve performance. *Parameters:* $D = 900$, $\eta_1 = 1$, $\eta_2 = 0$.

A common problem with REINFORCE is high variance gradient estimates leading to bad performance [Marbach and Tsitsiklis, 2003, Schulman et al., 2015]. The reward ($\eta_1$) and punishment ($\eta_2$) magnitude alters the variance of the updates, and we show that the interplay between reward, penalty and reward-condition and their effect on performance can be probed within our model. This framework opens the possibility for studying phase transitions between learning regimes [Gamarnik et al., 2022].

## 2.5 Speed-accuracy trade-off

Figure 5c shows the evolution of normalised overlap $\rho = R/\sqrt{Q}$ between the student and teacher obtained from simulations and from solving the ODEs in the case where $n$ or more decisions must be correctly made in an episode of length $T = 13$ in order to receive a reward (with $\eta_2 = 0$). We observe a speed-accuracy trade-off, where decreasing $n$ increases the initial speed of learning but leads to worse asymptotic performance; this alleviates the initial plateau in learning seen previously in fig. 5b at the cost of good generalisation. In essence, a lax reward function is probabilistically more achievable early in learning; but it rewards some fraction of incorrect decisions, leading to lower asymptotic accuracy. By contrast a stringent reward function slows learning but eventually produces a highly aligned student. For a given MDP, it is known that arbitrary shaping applied to the reward function will change the optimal policy (reduce asymptotic performance) [Ng et al., 1999]. Empirically, reward shaping has been shown to speed up learning and help overcome difficult exploration problems [Gullapalli and Barto, 1992]. Reconciling these results with the phenomena observed in our setting is an interesting avenue for future work.

# 3 Experiments

To verify that our theoretical framework captures qualitative features of more general settings, we train agents from pixels on the Procgen [Cobbe et al., 2019] game 'Bossfight' (example frame, fig. 6a (top)). To remain close to our theoretical setting, we consider a modified version of the game where the agent cannot defeat the enemy and wins only if it survives for a given duration $T$. On each timestep the agent has the binary choice of moving left/right and aims to dodge incoming projectiles. We give the agent $h$ lives, where the agent loses a life if struck by a projectile and continues an episode if it has lives remaining. This reward structure reflects the sparse reward setup from our theory and is analogous to requiring $n$ out of $T$ decisions to be correct within an episode. We further add asteroids at the left and right boundaries of the playing field which destroy the agent on contact, such that the agent cannot hide in the corners. Observations, shown in fig. 6a (bottom), are centred on the agent and downsampled to size $35 \times 64$ with three colour channels, yielding a 6720 dimensional input. The pixels corresponding to the agent are set to zero since these otherwise act as near-constant bias inputs not present in our model. The agent is endowed with a shallow policy network with

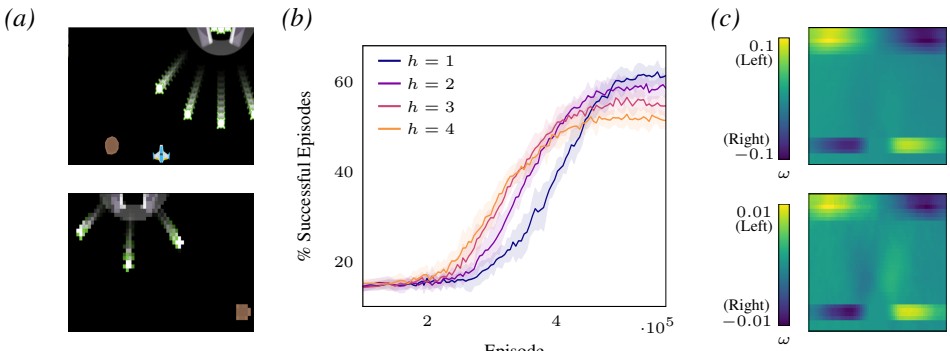

Figure 6: **Empirical speed-accuracy tradeoff in Bossfight.** *(a)* Top: Screenshot from a frame of 'Bossfight.' Bottom: Example observation provided to the agent's policy network. In our variant, the agent can move left or right and aims to survive for a given duration $T$. Collision with projectiles or asteroids costs one life, and the agent has $h$ lives before an episode terminates. *(b)* Performance during training, measured on evaluation episodes with $h = 3$ lives. Agents trained in stringent conditions ($h = 1$) learn slowly but eventually outperform agents trained in lax conditions ($h = 4$), an instance of the speed-accuracy tradeoff. Shaded regions indicate SEM over 10 repetitions. *(c)* Policy network weights for an agent with (top) $h = 4$ lives and (bottom) $h = 1$ life. For simplicity, one colour channel (red) is shown. Training with fewer lives increases the weight placed on dodging projectiles (see text). *Parameters: $T = 100, \eta_1 = 8.2e - 5, \eta_2 = 0$.*

logistic output unit that indicates the probability of left or right action. The weights of the policy network are trained using the policy gradient update of eq. (1) under a pure random policy.

To study the speed-accuracy trade-off, we train agents with different numbers of lives. As seen in fig. 6b, we observe a clear speed-accuracy trade-off mediated by agent health consistent with our theoretical findings (c.f. fig. 3c). Figure 6c shows the final policy weights for agents trained with $h = 1$ and $h = 4$. These show interpretable structure, roughly split into thirds vertically: the weights in the top third detect the position of the boss and centre the agent beneath it; this causes projectiles to arrive vertically rather than obliquely, making them easier to dodge. The weights in the middle third dodge projectiles. Finally, the weights in the bottom third avoid asteroids near the agent. Notably, the agent trained in the more stringent reward condition ($h = 1$) places greater weight on dodging projectiles, showing the qualitative impact of reward on learned policy. Hence similar qualitative phenomena as in our theoretical model can arise in more general settings.

## 4 Concluding perspectives

The RL perceptron provides a framework to investigate high-dimensional policy gradient learning in RL for a range of plausible sparse reward structures. We derive closed ODEs that capture the *average-case* learning dynamics in high-dimensional settings. The reduction of the high-dimensional learning dynamics to a low-dimensional set of differential equations permits a precise, quantitative analysis of learning behaviours: computing optimal hyper-parameter schedules, or tracing out phase diagrams of learnability. Our framework offers a starting point to explore additional settings that are closer to many real-world RL scenarios, such as those with conditional next states. Furthermore, the RL perceptron offers a means to study common training practices, including curricula; and more advanced algorithms, like actor-critic methods. We hope to extract more analytical insights from the ODEs, particularly on how initialization and learning rate influence an agent's learning regime. Our findings emphasize the intricate interplay of task, reward, architecture, and algorithm in modern RL systems.

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
