# OpenReview forum: "The RL Perceptron: Generalisation Dynamics of Policy Learning in High Dimensions"
_NeurIPS.cc/2023/Conference — Submitted to NeurIPS 2023_

### Official Review · Reviewer_PTjw · 2023-06-27

**Soundness:** 3 good
**Presentation:** 3 good
**Contribution:** 3 good
**Rating:** 6
**Confidence:** 4

**Summary:**

This paper focuses on developing theories for the learning dynamics of policy gradient reinforcement learning (RL) algorithms with a particular focus on high-dimensional latent feature space. As an early work along this direction, the authors study a binary-action environment setup for simplicity. The authors develop ODE-based learning dynamic equations that generalizes across diverse protocols, including different policy horizons, the existance of failure penalties, and the choices of dense / sparse rewards. The authors further develop theories for optimal learning rate, optimal horizon scheduling, and learnability with respect to these hyperparameters. The authors finally conduct an experiment on vision-based Procgen Bossfight environment and demonstrate that under more general settings, similar phenomena arise as their theoretical model developed under simpler setups.

**Strengths:**

Overall the paper is well-structured and well-written. Under the binary-action environment setup, the ODE dynamics model developed by the authors is shown to accurately describe diverse common learning setups. There are also extensive experiments and plots that illustrate the difference of learning dynamics under different environment and optimization parameters, which provide much insights for the readers.

**Weaknesses:**

In the main paper, the latent feature space dimension $D$ is fixed to 900 except Procgen. It would be helpful if authors provide more analysis on the influence of latent dimension $D$ on the learning dynamics. Empirically, for which $D$ is author's proposed ODE-based learning dynamics equation still accurate?

For Fig. 6b, Plotting environments of different episode lengths by comparing their "Number of episodes" seems misleading. Authors claim that agents learn slower for environments with shorter episode lengths, but this is not accurate. If one compares the total time step of learning (num episodes * episode length), agents acturally learn faster on these shorter horizon environments.


**Questions:**

See "weakness" section above.

Minor typos:

- Equation 2: looks like 1-eps_g is defined as disagreement instead of eps_g.

- Line 194, Fig. 3c -> Fig. 2c

**Limitations:**

Limitations need to be explicitly addressed in the conclusion section, including (1) the simplicity of the problem setup studied by the paper (binary action scenarios); and (2) the paper's focus on shallow, one-layer neural network that takes high dimensional feature as input. For more general applications, neural networks typically consist of many layers, activations, and normalizations stacked on top of each other, so empirical analysis in these scenarios will be particularly helpful.

---

> ### Author Rebuttal · Authors · 2023-08-09
>
> We thank you for the time you took to review our paper, and appreciate the comments raised, which will help to improve the quality of the paper. We address these comments below:
>
> * **Data space dimensions**: The accuracy of the ODE description of the learning dynamics hinges on the concentration of the order parameters – i.e how quickly the sums in eq 3 converge to their mean (by the central limit theorem these sums converge to their mean with probability 1 in the limit $D \rightarrow \infty$, which makes it possible to map the stochastic evolution of the student onto a deterministic evolution of the order parameters). These sums have variance  $\mathcal{O}(1/D)$, so we see that latent feature space dimension $D$ controls the size of the stochastic fluctuations of the update to the student vector/relevant overlaps. In decreasing $D$, fluctuations would  become more influential, causing increasing deviation of the stochastic evolution of the student from the ODEs over the course of learning. $D=900$, was chosen as a suitably high dimension to conduct simulations to ensure stochastic fluctuations would not have too great an effect. We have also performed simulations with $D=400$, which in general produces good agreement with the ODEs, but in some particular learning scenarios there is visible deviation when viewed on a logarithmic scale. A visualisation of this is provided in fig. C of the attachment.
>
> * **Speed and accuracy in Fig 6b**: We should have been clearer on exactly how an episode ends. In the Bossfight game every episode is pre-set to 100 timesteps. Within an episode each time the agent hits a projectile, it incurs a penalty of 1. At the end of an episode if the total penalty is $\geq$ number of lives, then the update is not carried out. So in this particular set-up, the episode duration is equal for all 4 agents with different lives. This mirrors our theory, and reflects a real world setting where ‘success’ can only be judged at the end of an episode, but is determined by decisions taken along the way. However, we agree that in other training scenarios an episode would terminate at the moment all lives are lost, meaning that for fewer lives episode lengths would tend to be much shorter in the early instances of training, and using the metric (episode length * no. of episodes) would be more appropriate for measuring real-time duration. We can treat this case in our theory and will add its behaviour in the revision. We emphasise that we are comparing environments with different lives but fixed duration, and claim that environments with more lives learn faster (but to a lower asymptotic accuracy); we do not ‘claim that agents learn slower for environments with shorter episode length’.
>
> A related realistic training scenario is presented in figure 10 in the submission (or neater in fig. D in the attachment). Here we employ the full REINFORCE policy gradient update for learning PONG and do indeed plot and compare environments of differing maximum episode length on the same x-axis (number of episodes completed). We observe that agents trained on shorter episode lengths initially learn much faster, but reach a lower asymptotic accuracy (we think there might be a misunderstanding, as this is the opposite of what you write that we claim, but please correct us if we’re wrong). In this case, the episode does indeed terminate when the agent dies, and we agree it may be more appropriate to measure real time duration as (episode length * no. of episodes), which we will change in the revision; however in this case it would not change the ordering for speed of learning
>
> Minors:
> * We thank you for reading thoroughly enough to indicate our typos
>
> ## Limitations:
> We agree with the reviewer on the need for a dedicated section explaining the limitations of our model, which are currently scattered around the paper. We will discuss that the model is only able to capture binary action spaces; that the environment data is being sampled from a Gaussian distribution, which is an idealised setting; our focus on shallow networks; and the fact that we do not consider state transitions conditional on the action chosen, which is something we are looking to include in future iterations of our model.

---

> > ### Comment · Reviewer_PTjw · 2023-08-10
> > **Reply**
> >
> > Thanks for authors' rebuttal! My concerns (different latent space dimensions and Fig 6b experiment clarification) have been sufficiently addressed and I'll increase my confidence.
> >
> > (For "slower" in my original comments I meant a slower rate of asymptotic convergence, not the initial learning speed)

---

> > > ### Author Response · Authors · 2023-08-18
> > >
> > > We’re glad to have been able to address the reviewer’s concerns.
> > >
> > > * **Additional experiments**: We thought it also worth mentioning that we do indeed carry out empirical analysis on a learning scenario with a deep non-linear network. We forgot to mention in the original rebuttal and the reviewer pointed out as a limitation of the paper.
> > > In the results on Atari “Pong” (fig. 10 in the supplementary and fig.D in the rebuttal PDF), we used a deep convolutional neural network for the policy, specifically 2 convolutional layers and 2 fully connected layers with ReLU non-linearities trained with the Adam optimizer.
> > >
> > > * **Slower rate of asymptotic convergence**: Apologies for the confusion. We actually do not make claims on the ‘rate of asymptotic convergence’, but we refer to asymptotic expected reward achieved by the agent. In line 267 - ‘We observe a speed-accuracy trade-off, where decreasing $n$ increases the initial speed of learning but leads to worse asymptotic performance’. We hope this clarifies.
> > >
> > > We again thank the reviewer for acknowledging our fulfilment in addressing their concerns. In light of this, we wonder if the reviewer would consider **a score adjustment**, too.

---

### Official Review · Reviewer_9GSU · 2023-07-05

**Soundness:** 3 good
**Presentation:** 3 good
**Contribution:** 3 good
**Rating:** 6
**Confidence:** 4

**Summary:**

This paper proposes a model for solving high-dimensional problems in reinforcement learning (RL) referred to as the RL perceptron. The model is used as a framework for analyzing generalization dynamics of simple neural networks for RL tasks. More precisely, the model employs a student teacher design in which the student takes a sequence of choices and the correct choices are given via the teacher. However, the student does not have access to the correct choice at every step but rather only receives a signal at the end of each episode. As such, the model is studied as a sequential version of the perceptron algorithm. The work first derives a set of differential equations that capture the learning dynamics of the model. The dynamics are analyzed via the overlap of the weight vectors of the student and teacher respectively. The manuscript studies multiple reward settings: a vanilla setting that constitutes sparse rewards, a setting with penalty at every step and a setting where the agent is given small sub-rewards after a certain amount of time. Each of the proposed settings provides insights into how different reward functions lead to different solutions for optimal parameters when solving the system. The derivation of optimal hyperparameters demonstrates that annealing the learning rate and building a curriculum of episodes are crucial for optimal convergence. Then, it is shown that there exist phase transitions under different learning rates that can lead to convergence to sub-optimal minima and that there is a speed-accuracy tradeoff when varying reward functions. Lastly an experimental section provides insides in the practical properties of the algorithm that closely follow the analytical results on the speed-accuracy tradeoff.

**Strengths:**

1.) First, I would like to say that I enjoyed reading this paper. I think it is well written and well structured with a clear line of reasoning throughout the manuscript. The figures are illustrative of the analysis and the captions are sufficiently descriptive to understand the plots quickly.

2.) The idea of employing a perceptron-like algorithm in order to understand the policy gradient system dynamics is novel to the best of my knowledge and the generated insights are interesting. I think this is a nice contribution as it offers a way to analyze high-dimensional RL systems in a different way than the commonly employed linear MDPs.

3.) The manuscript is technically sound and the analysis provides a good understanding of the proposed algorithm and its inner workings. I did check the math in the appendix for crude errors and was unable to find any but I did not try to understand all the math in detail.

4.) The manuscript provides a way to think about RL systems that is not common and I think it is likely going to be useful in understanding some parts of the systems that may have been hard to understand previously. As such, I think it is a decent contribution that can likely be built upon by others in the sub-field of policy gradient methods.


**Weaknesses:**

a.) The connection to policy gradients was not immediately clear to me. It might make sense to move equation 9 in place of equation 1 and highlight the connection between the perceptron update rule and the REINFORCE algorithm in a brief sentence. I think it would be good to highlight that the update rule uses an approximation that is only accurate early in training.

b.) The limitations of the model are addressed rather sparsely. I think the work would benefit from having a clearer picture of the weaknesses of the approach which would enable researchers to use it and improve upon it in the future.

c.)  The model seems to be unable to solve the benchmark problem fully. However, it is hard to tell whether that is just a limitation of the model or whether the task is hard the way it is designed with the changes in the manuscript. Having a baseline performance line in the plot or giving a brief sentence of what the expected performance of a commonly used RL algorithm on the benchmark is would be very useful to determine the capabilities of the method.

d.) I think one key thing that is missing from the paper and would make it a very strong contribution is to show the relationship between the proposed model and common deep RL methods. It would have been nice to have a direct comparison from the proposed method to a neural network approach using standard REINFORCE-like updates to see if there is a correlation produced from insights of the RL-perceptron with the behavior of the regular deep neural network. I do understand that space is limited though.

----

Overall, I think changes that could improve the manuscript would establish the model's connections to other research that people have done in the area. This could, for instance, include baseline performances, transfer of insights to other methods or any theoretical results that put the work into reference with commonly knows results.

Minor clarity suggestions:
* Line 250, there is a broken off sentence in there that should be removed.
* Line 562 equation reference is missing.


**Questions:**

Q1: What is the relationship between the proposed algorithm and standard policy gradient approaches with deep neural networks? Are there any conclusions that can be transferred from the RL perceptron to standard deep RL algorithms?

Q2: What is the expected performance on the benchmark task where we would consider the task solved? Is it possible to win 100% of the time? This seems to be crucial information in order to determine the practical utility of the algorithm.

Q3: One question that I have is about the connection between the proposed method and the standard perceptron algorithm. Is there a reduction from the RL perceptron to the standard perceptron?

Suppose the model is updated at the end of every epoch. I’m thinking something along the following lines. Let n be the number of correct guesses made by the algorithm and N be the number of total timesteps in a trajectory. Then N-n is the number of incorrect guesses. Assume the sparse reward model where we have to be correct on k guesses to see a reward. Now, take a full trajectory as an input to the algorithm. If k < n, label the trajectory positive, otherwise negative. Of course there is some detail missing that would include the teacher in the labeling process but you should get a rough idea. If such a reduction exists it would reduce the problem to binary classification of whether or not a certain action sequence will lead to success.


**Limitations:**

I do not believe there is any negative societal impact that needs to be addressed regarding this work. The limitations are addressed rather sparsely. As stated before, I believe that the manuscript would benefit from more structured limitations sections. One limitation that I see is that the model currently requires actions to be discrete and (possibly ?) binary. While for a first version of the model this is absolutely fine, this might be something that researchers can work on in the future. Another limitation of the model is that it seems to not be able to fully solve the suggested benchmark problems. Again, I don't believe this to be an issue for the manuscript as the goal is not to provide a state-of-the-art model but rather to make progress towards understanding the learning dynamics of deep RL systems. Yet, in the future a goal should be to have models that can be described analytically that also achieve comparably high performance on realistic tasks.

---

> ### Author Rebuttal · Authors · 2023-08-09
>
> We thank you for the time spent reviewing the paper and for the positive comments, and we're glad you enjoyed reading the paper. We really appreciate the comprehensive feedback.
>
> Below, we go through the points made and address the questions after.
>
> * **Connection to policy gradient**: We agree with moving eq. 9 of the appendix to the main text. As noted by the reviewer and stated in the derivation of the update rule, our approximation is formally valid in the early stages of training. However we believe that this is an important phase of learning and that its study complements recent work focusing on the convergence of policy gradient methods. By analysing the early learning dynamics, we are able to capture effects like the speed-accuracy trade-off which also appears in our experiments in more realistic settings using the full REINFORCE policy gradient algorithm (Bossfight and Pong fig. A and fig. D of the attachment respectively)
>
>
> * **Limitations**: We agree also with this point. We plan to collect the limitations in a dedicated paragraph in the conclusions, collecting the comments on the limitations and expanding further to include possible extension of the model. The limitations would include: the model only being able to capture binary action spaces, the environment data being sampled from idealistic Gaussian distributions, we focus only on shallow networks, and we do not consider state transitions conditional on action, which is something we are looking to address.
>
>
> * **Model ‘performance’**: While the model performance on the benchmark problem is decent (over 60% final episode success rate), deep RL algorithms will certainly do significantly better. The point we are trying to make here is not that the model we analyse performs particularly well in terms of performance, but rather that the phenomena observed in the experiment are captured by the teacher-student setup we study theoretically (e.g. speed-accuracy trade-off in Fig 6b). Indeed, making these comparisons is only interesting if the model performs reasonably well on the benchmark, a criterion we feel is more than satisfied. We emphasise that we are not claiming ‘practical utility’ of our model, and as you mention ‘our goal is not to provide a state-of-the-art model but rather to make progress towards understanding the learning dynamics of deep RL systems’. We of course agree with your point that the future goal would be to be able to describe models analytically that can also achieve comparably high accuracy on realistic tasks, but starting with an analysis of REINFORCE policy-gradient is a prerequisite first step in this direction. We have preliminary results for full REINFORCE policy-gradient on a shallow network for bossfight (fig. A of the attachment) which you may consider a ‘benchmark’ for comparison.
>
>
> * **Deep architectures**: We have some results in the appendix for deep architecture that we will move up to the main text should our paper be accepted, which would grant us an additional page. We considered Atari “Pong” using policy-gradient on a deep network (two convolutional and three fully-connected layers with ReLU nonlinearity and ADAM optimiser) and observed a similar behaviour as in our simple model -- in particular, the deep model also shows a speed-accuracy trade-off (Fig. D of the attachment). We also have preliminary Bossfight results, as mentioned at the end of the last answer. In terms of comparing to other common deep RL methods, we thought best to focus initially on pure REINFORCE policy gradient (and not its variations), but we believe our methods can be extended.
>
> Thanks for the minor comments, we will correct those in a revised version.
>
> ## Questions
>
>
> * **Transferable conclusions**: This is a very interesting question that we would like to investigate further. Based on the result of our early experiments with deeper architectures on “Bossfight” in Sec.3 and Atari “Pong” in the appendix (see our answer above), we do believe that our simple model captures some aspects of training deeper neural networks with policy gradient methods. However, understanding exactly under what conditions our results break, is something that we are keen to explore in future works by adding additional elements to our model, such as larger action spaces. In particular we are interested in understanding what happens for more complex state spaces and richer transition dynamics.
>
> * **Regarding performance**: Please see our reply on “model performance” above.
>
>
> * **Perceptron Algorithm**: This is an interesting question - the key difference between the standard perceptron learning algorithm and REINFORCE is that the former only considers one input at a time, and updates the weights of the student when the student makes an error; REINFORCE updates the student’s weights instead when it correctly predicts the label of all (or of a finite fraction of) the inputs in an episode. The perceptron algorithm is thus in some sense the opposite of REINFORCE. This difference is important; for example, the speed-accuracy trade-off is specific to REINFORCE.

---

> > ### Comment · Reviewer_9GSU · 2023-08-12
> >
> > Dear authors, thank you for the clarifications and the detailed rebuttal. I’m glad that you found some of my feedback helpful.
> >
> > **Model Performance**
> > Thank you for highlighting this. I wanted to make a brief clarification here. I understand that the goal is not to derive a novel algorithm. However, any theoretical framework that ought to explain the behavior of a class of models is arguably most useful if its insights correlate with methods of the class we actually employ. I’m not claiming that I would expect your method to outperform any state-of-the-art algorithm but comparisons like this show us the magnitude of unexplained from the theoretical model. I did not have to pick performance but could have picked several other metrics. “Capabilities” in my review was meant as capabilities of the model to explain real-world methods. That is why I was highlighting this point in particular.
> >
> > **Reduction to Perceptron** (Note, I don’t think this point is going to change my perception of the paper, I just think it’s interesting to discuss)
> > I understand the framing of the model as a perceptron algorithm on a sequence. However, I can easily define a single sequence to be “an input” and then the RL-perceptron also only considers one input at a time. Next, could re-label any sequence with a single label, true or false if condition met. Whether I label “predicted correctly” as true or as false does not matter to the algorithm. I feel like I might be missing something here.
> >
> > **I have also read through the other reviews.** I agree with many of the points made. I would like to highlight that I *do* believe the experiments are sufficient to support the claims and experiments on multiple environments have been conducted.
> >
> > Overall, I maintain that this paper should be accepted since I think it is an insightful piece of work that will be beneficial to the community. Since it is not clear to me that this approach will in the future explain various, intricate behaviors of commonly employed methods we use, I will retain my score.

---

> > > ### Author Response · Authors · 2023-08-18
> > >
> > > We thank you for your continued engagement, and are glad to hear you think our paper will be beneficial to the community. We’re happy to further discuss the interesting points you raised, below:
> > > * **Model Performance:** Thank you for your clarification, now understanding that you used ‘performance’ as an example of a metric used to compare to real-world methods (specifically the REINFORCE policy gradient method), we would like to point you to figures A and B of the attachment. Here a comparison (of learning dynamics) can be made of the full REINFORCE policy gradient update using a shallow network input to a sigmoidal policy (fig. A) to our RL-perceptron update (fig. B) on the same Bossfight environment. We agree that this comparison is beneficial and is something we would like to include in the revised version of the paper, should it be accepted. We agree that there remains a sizeable gap to bridge from our model to more realistic performances etc. Indeed, we have some concrete ideas that should help closing this gap: extending the model to problems with more states and more decisions using results from statistical physics [10]; from there, defining a notion of ‘value’ on states/actions, offering the possibility of incorporating value-based RL algorithms or algorithms that combine policy and value-based methods (like the actor-critic); considering higher-dimensional action spaces; etc. We will discuss these avenues for further work in the revised manuscript.
> > >
> > > * **Reduction to Perceptron:** We agree this is an interesting point for discussion.
> > >     In the case of **undiscounted sparse rewards** (reward received only at the end of an episode dependent on successful completion), it would be possible to label an entire sequence as true/false. However, this would require the student to be involved in the labelling process, and would also require the tying together of weights because the same student weights are applied at each time step (if the sequence was concatenated in the process of classification).
> > >     As mentioned, this would only be possible for the case of undiscounted sparse rewards. If rewards were additionally received within the episode (e.g. receiving a small reward after every correct decision) and/or with a discount factor of $\gamma$ (explained below) then the reduction to a binary classification as a perceptron would not be possible. This extension only requires a simple adjustment of our update: $\mathbf{w}^{\mu+1}  = \mathbf{w}^{\mu}+\frac{\eta}{\sqrt{D}}\left(\frac{1}{T} \sum_{t=1}^{T}y_t  \mathbf{x}_{t}  G_t\right)^\mu$,
> > >
> > >     where $G_t = \sum_{t^\prime = t}^T \gamma^{t^\prime-t}R_{t^\prime}$ is the total discounted reward from time $t$, $\gamma \in \left(0,1\right]$ is the discount factor and $\eta$ is the learning rate as before. We have since extended our approach to be able to incorporate this more general form.
> > >
> > >     The binary classification reduction would also not be possible if we were to extend to multiple distributions with, and action dependent state transitions (as then of course a single sequence will need to be generated with the student), which is something we are looking to extend to.

---

### Official Review · Reviewer_nMd7 · 2023-07-06

**Soundness:** 2 fair
**Presentation:** 2 fair
**Contribution:** 2 fair
**Rating:** 6
**Confidence:** 1

**Summary:**

The authors develop a set of differential equations that describe the learning dynamics in high-dimensional settings, allowing for a quantitative analysis of learning behaviors. This framework enables the computation of optimal hyper-parameter schedules and the visualization of phase diagrams for learnability. It also serves as a starting point for exploring RL scenarios closer to real-world situations, including those with conditional next states. The RL perceptron can be used to investigate various training practices, such as curricula, and advanced algorithms like actor-critic methods. The authors aim to gain analytical insights from the differential equations, particularly regarding how initialization and learning rate affect an agent's learning process. Overall, their research highlights the complex interplay between task, reward, architecture, and algorithm in modern RL systems.

**Strengths:**

* The paper tackles an important problem of understanding high dimensional RL policies.
* The method section seems theoritically sound.

**Weaknesses:**

Experiments are not convincing. For a paper investigating such an important problem, the paper should have shown results on multiple environments.

**Questions:**

I don't have any questions.

**Limitations:**

A considerable limitation of the paper is the lack of a comprehensive evaluation of the method. The paper provides only results on one experiment. I would suggest the authors conduct more thourough experiments in order to make the applicability of method more clear.

---

> ### Author Rebuttal · Authors · 2023-08-09
>
> We thank the reviewer for providing a review of our paper. However, we have to disagree on the two points raised by the review:
>
> We thank the reviewer for providing a review of our paper. However, we have to disagree on the two points raised by the review:
>
> * We want to stress that our paper does not propose a method; **our goal is  to analyse the dynamics of the REINFORCE policy gradient algorithm**, perhaps the simplest algorithm for reinforcement learning. For example, we would like to know: what are good choices for the learning rate or the episode length? How quickly is my model going to learn? How well is it going to generalise at the end? And this is studied through our analytically solvable framework.
>
>
> * Contrary to what the reviewer claims, **we did conduct experiments in different environments** to verify the predictions of our theory: we consider the environment “Bossfight” with a shallow architecture (fig. 6b) and Atari “Pong” (fig. 10 in the appendix) with deep network architecture (we will be moving this experiment to the main body thanks to additional space should the paper be accepted).
>
>
> Should the reviewer have any questions during the discussion period, we would be more than happy to answer them.

---

> ### Comment · Area_Chair_Ek2d · 2023-08-16
> **Reveiwer Response Needed**
>
> Hello Reviewer,
>
> The authors have made efforts to address your comments on their work via the rebuttal. Part of the NeurIPS review process is participating meaningfully in the rebuttal phase to help ensure quality. Please read and respond to the author's comments today, latest tomorrow, to give everyone time to respond and reach proper conclusions.
>
> Thank you for your assistance in making NeurIPS a great conference for our community.

---

### Official Review · Reviewer_vS6U · 2023-07-06

**Soundness:** 2 fair
**Presentation:** 1 poor
**Contribution:** 3 good
**Rating:** 6
**Confidence:** 2

**Summary:**

The work proposes a theoretical framework to study the average case learning behavior of deep RL policy gradient methods. The framework is based on ODEs that can describe the typical learning dynamics of PG RL agents. The framework is used in various settings to describe learning behaviors in these and a final experiment on training a policy-gradient agent in a ProcGen game is added to bridge the theory-practice gap. This final experiment verifies that a speed-accuracy trade-off exists in practice, similar as predicted by the theoretical framework.

**Strengths:**

* The work addresses an important aspect in deep RL research as most theoretical guarantees for RL are not well connected to the practical side of deep RL.
* The work sets out to provide a theoretical framework from which to study deep RL methods.
* It shows how such the theoretical framework can be used to understand the influence of
  * delayed rewards and reward penalties
  * learning rate schedules and episode lengths
  * reward stringency

**Weaknesses:**

* The work was very difficult to follow for me. Due to the structure of the paper, many aspects seem to "fall out of thin air".
* Without Appendix A it seems impossible to begin to understand Section 2 since more assumptions about the student teacher environment are given.
* It often feels like the work requires extensive prior knowledge to be understandable.
* Wording is often confusing. For example, in the beginning when talking about the reward for the RL-Perceptron case Fig 1 has a description about rewards that seems permissible for very dense rewards, whereas lines 48-50 talk only about extremely sparse rewards.

Overall the paper seems very interesting and full of great ideas but due to a somewhat convoluted presentation and missing details that seem to be pushed to the appendix it falls short of clearly communicating these ideas.

I might have missed something obvious, but to me it seems that the paper would first need fairly substantial rewriting to be easier to parse before it can be accepted.

**Questions:**

Could the framework be adapted to work for other RL agent types?

**Limitations:**

The authors do not explicitly list limitations of their framework

---

> ### Author Rebuttal · Authors · 2023-08-09
>
> We thank you for the time spent reviewing, and appreciate the feedback given. We understand clarity is something we can work on, and will strive to order definitions and assumptions in a way that is easier to follow in the revised version. Below we respond to the points you raised, and we would be grateful for any questions to help with further clarification.
>
> * **Results for derivation/explanation / assumptions in appendix / prior knowledge needed**: Unfortunately, the page limit forced us to push parts of the derivations to the appendix; we tried to follow the usual convention adopted in other papers on neural network theory published at NeurIPS recently, cf. refs [4-7]. We are happy to rethink how to divide results between main text and appendix; to that end, can we ask the reviewer to identify some concrete points that could be clarified? We will then try to improve those for the next version of the manuscript. Our work does make extensive use of methods from statistical physics, but we believe clarity is paramount and have endeavoured to make the technical presentation in the supplement self-contained.
>
> We plan to move to the main text the assumptions from appendices B concerning the early learning time equivalence to the REINFORCE policy gradient update for a softmax policy. In particular, we are considering (i) presenting equation 9 in appendix B along-side equation 1 in the main body (ii) clarifying the input distribution in section 2.1 (iii) adding detail on the structure of the state space in our model and on a probabilistic policy, which we emphasise is only there to establish the connection to the reinforce policy gradient algorithm.
>
> Does the reviewer think that these changes will improve the clarity of the paper without adding details irrelevant to the understanding of the main text? Are there other concepts that require clarification? In case there are some elements that the reviewer thinks are still missing, please do let us know.
>
>
> * **Regarding reward density**: In our setup we mainly focus on the situation where rewards are only received at the end of an episode. Further, in fig 1b, the diagram shows that $\Phi$ (which controls the reward signal) is only calculated once all rewards are received. We may perhaps be using a different definition of reward sparsity; we characterise an environment with a sparse reward as one in which a meagre amount of state-action pairs in a trajectory return a feedback signal (reward). May we ask the reviewer to clarify where our text might suggest dense rewards? We would like to clarify this issue in the text.
>
>     We agree that **capturing dense rewards** would be beneficial to extending the generality of our model. This extension only requires a simple adjustment of our update: $\mathbf{w}^{\mu+1}  = \mathbf{w}^{\mu}+\frac{\eta}{\sqrt{D}}\left(\frac{1}{T} \sum_{t=1}^{T}y_t  \mathbf{x}_{t}  G_t\right)^\mu$,
>
>     where $G_t = \sum_{t^\prime = t}^T \gamma^{t^\prime-t}R_{t^\prime}$ is the total discounted reward from time $t$, $\gamma \in \left(0,1\right]$ is the discount factor and $\eta$ is the learning rate as before. We have since extended our approach to be able to incorporate this more general form. However, we stress that the key ingredient to capture relevant aspects of high-dimensional RL is the presence of sparse reward in the first phase of learning.
>
> ## Questions
> * **‘Could the framework be adapted to work for other RL agent types?’**: yes, and we are working on it. There are still limitations (coming to the point of “Limitations”) which at the moment are scattered in the paper. In summary, at the moment we can only analyse REINFORCE policy gradient. It is possible to extend the model by considering more states with a variety of data distributions, following for example Ref. [10]. Using this it would be possible to define the notion of ‘value’ on states/actions, meaning there is the potential to incorporate value based RL algorithms or algorithms that combine policy and value based methods (like the actor-critic), and this is a plan for future works. It would also be possible to extend to multi-state action spaces instead of binary by considering work that finds learning curves for the multi-class perceptron [8]. This again would widen the possibilities of RL agents we can consider, and also widen the number of suitable environments we can test against.
>
> ## Limitations:
> We did indeed not address limitations in a separate, self-contained section in the paper, and instead scattered the limitations throughout the paper. We will add one paragraph to the conclusion in the revised version. The limitations would include: our model currently only captures binary action spaces, the environment data being sampled from a Gaussian distributions, which is an idealised setting; we focus only on shallow networks, and we do not consider state transitions conditional on action, which are all things we are looking to address in future versions of the model.

---

> > ### Comment · Reviewer_vS6U · 2023-08-11
> > **Rebuttal Response**
> >
> > Thank you for the detailed rebuttal. I believe you address all of my concerns.
> >
> > ### Regarding Presentation
> > > We plan to move to the main text the assumptions from appendices B [...]. Does the reviewer think that these changes will improve the clarity of the paper without adding details irrelevant to the understanding of the main text?
> >
> > Yes absolutely! I believe that would have helped me a lot when reading the paper the first time. I agree that not everything needs to be in the main part of the paper, but it only started to click for me after I had a look at appendix B.
> >
> > ### Regarding reward density
> > > May we ask the reviewer to clarify where our text might suggest dense rewards? We would like to clarify this issue in the text.
> >
> > I apologize. But it seems I was a particularly dense reviewer. As written in my original review, Lines 48 - 50 caused my confusion.
> > The first part states "[... The student] receives a reward which depends on whether earlier decisions are correct. [...]" I misinterpreted this statement as meaning a reward for every decision. The second part then clearly states that the sparse reward is considered which is also confirmed by the caption of Figure 1. I didn't do a good job in communicating where my misunderstanding comes from and I apologize for that.
> >
> > ### Updating my score
> > I have read all other reviews and the rebuttals. Since my initial score deviated most substantially from all other reviewers I have also re-read the paper. I now do believe that my original score was too low. I still see that the work has substantial value to the community and, as stated originally, is full of great ideas. As such it should have only been, at worst a weak reject or borderline accept. Having already stated that I believe the rebuttal to address my concerns, I do increase my score.

---

### Official Review · Reviewer_bMDj · 2023-07-17

**Soundness:** 4 excellent
**Presentation:** 3 good
**Contribution:** 4 excellent
**Rating:** 7
**Confidence:** 3

**Summary:**

The paper discusses the application and theoretical understanding of Reinforcement Learning (RL) algorithms in high-dimensional settings. The authors propose a high-dimensional model of RL that can capture a variety of learning protocols and derive its dynamics as a set of closed-form ordinary differential equations.

The authors introduce the RL perceptron, a model for high-dimensional, sequential policy learning. In this model, a student network learns from a teacher network in a sequential decision-making task. The student does not observe the correct choice for each input; instead, it receives a reward that depends on whether earlier decisions are correct.

The authors derive an asymptotically exact set of ODEs that describe the typical learning dynamics of policy gradient RL agents. They use these ODEs to characterize learning behavior in a diverse range of scenarios, including exploring several sparse delayed reward schemes, deriving optimal learning rate schedules and episode length curricula, identifying ranges of learning rates for which learning is 'easy,' and 'hybrid-hard,' and identifying a speed-accuracy trade-off driven by reward stringency.

They also demonstrate that a similar speed-accuracy trade-off exists in simulations of high-dimensional policy learning from pixels using the procgen environment "Bossfight" and Atari "Pong". The authors aim to close the gap between theory and practice in high-dimensional RL.
The paper also discusses the sample complexity in RL, statistical learning theory for RL, and dynamics of learning, providing a comprehensive overview of the current state of RL theory and practice.

**Strengths:**

This is a robust theoretical paper that enhances our comprehension of high-dimensional RL policy learning. The paper's claims are supported by high-level experimental evidence. A significant advantage of this paper is its use of the challenging procgen "Bossfight" environment, which is much closer to the real use cases compared to the preceding works. Additionally, the "Pong" game was analyzed in the supplementary materials to demonstrate the speed-accuracy tradeoff. Another strong aspect of the paper, and a significant advantage, is its focus on analyzing the average-case scenarios rather than the worst-case ones. The authors have also released the code to reproduce the results, further strengthening the paper's credibility.

**Weaknesses:**

To the best of my knowledge, the paper does not exhibit any significant weaknesses.

**Questions:**

1) Do you expect any substantial changes in the main results and conclusions in the paper when bigger/deeper policies are used?
2) What about environments with higher-dimensional action space?

**Limitations:**

Limitations were addressed reasonably well.

---

> ### Author Rebuttal · Authors · 2023-08-09
>
> We thank you for the time spent reviewing and the positive comments on our work, we really appreciate it. We’re glad you acknowledged the value of analysing scenarios in average-case instead of worst-case.
>
> We thank you for your questions, they are indeed ones which we have been considering ourselves for further extension/reinforcement of our work. We answer them below.
>
> ## Questions
>
> * **Bigger/deeper policies**: We did consider deep architectures empirically in our experiments on Atari "Pong", where we trained a DNN with 2 convolutional layers and 2 fully connected layers with ReLU non-linearities in-between. We found results consistent with the speed-accuracy tradeoff,  (Fig. 10 of the submission supplement, reproduced and cleaned up as fig. D of the attachment, this will be moved to the main text). We also verify the speed-accuracy trade-off on our modified bossfight game trained with the full REINFORCE policy gradient update using a shallow network (fig. A of the rebuttal pdf). Both experiments showed a speed-accuracy trade-off compatible with the one predicted by our model, which hints at a more universal phenomenon. We are currently testing the extent to which other results, such as learning rate schedules and the phase transition in the performance as a function of the reward rate, extend to deeper architectures. The goal of our approach, and this paper, is indeed finding a minimal model that captures several features of more complex RL problems. A future direction is to investigate the consequences of our analysis in applications, for example on learning rate and episode length schedules.
>
> * **More complex action space**: Being to our knowledge the first paper along this line, we wanted to consider the simplest setting. However, it is possible to extend our model to problems with more states and more decisions using results from statistical physics [10]. From there it would be possible to define the notion of ‘value’ on states/actions, meaning there is the potential to incorporate value-based RL algorithms or algorithms that combine policy and value-based methods (like the actor-critic), and this is a plan for future works. It would also be possible to extend to higher-dimensional action spaces instead of binary by considering work that finds learning curves for the multi-class perceptron [8]. This again would widen the possibilities of RL agents we can consider, and also widen the number of suitable environments we can test against.

---

> ### Comment · Area_Chair_Ek2d · 2023-08-18
> **Revwier Reponse Requested**
>
> Hello Reviewer,
>
> The authors have made efforts to address your comments on their work via the rebuttal. Part of the NeurIPS review process is participating meaningfully in the rebuttal phase to help ensure quality. Please read and respond to the author's comments today, latest tomorrow, to give everyone time to respond and reach proper conclusions.
>
> Thank you for your assistance in making NeurIPS a great conference for our community.
> -- Your AC

---

### Official Review · Reviewer_n4bo · 2023-07-19

**Soundness:** 2 fair
**Presentation:** 2 fair
**Contribution:** 2 fair
**Rating:** 5
**Confidence:** 3

**Summary:**

This paper introduces a new problem definition and class of solution methods for decision making.


The motivation put forward by the authors is that the current pool of solution methods do not have theoretical results that capture the neural functional classes used by practical applications in RL, leaving a gap between theory and practice.

The authors claim that the model of learning by reinforcement that they propose has sufficient flexibility to capture the same class of problems as the classic RL model, and that their problem definition, despite its generality, is solvable in closed form for higher dimensionalities (and even infinite), as opposed to the classic RL model, presumably formalised as POMDPs. In addition, they claim their model (unclear whether the authors mean the solution method or the problem, see detailed comments) behaves similarly w.r.t. hyperparameters, for which they provide optimal schedules and hypersensitivity plots.

Moreover, the authors claim their proposed class of problems and solution methods exhibits in practice similar behaviour as predicted by theory, and closes a gap between theory and practice, providing as evidence empirical illustrations in a particular environment, specially designed, called “Bossfight”, which they engineer using a platform for a procedurally generated environments.


**Strengths:**

$
\textbf{Originality}$

The authors propose a new problem definition and class of solution methods for RL, particularly policy optimisation. They describe their model as a perceptron, which they call “the RL perceptron”, and analyse the learning dynamics of a particular solution method for action space of 2 discrete actions. They then define different problem instances via heuristics for various feedback signals that such a solution method can receive. The authors propose a new perspective on RL inspired by statistical mechanics and dynamical system theory.

$\textbf{Quality}$

The particular math and statistical expressions the authors derive was not checked in detail beyond  the first few simple update equations, which seem correct. The appendix was not verified for correctness.
Plots also appear to be consistent with the experimental claims provided in text.


$\textbf{Clarity}$

The paper is at times clear, with some mysterious redefinitions, naming, and confusions (see in the next sections). Although the flow is cursive, the authors do not use the general breakdown of the paper and miss important sections like a “Background” section, which is placed in the introduction. It does not bother much, but clearly marking such section can help the reader understand better what is novel and what is known.


$\textbf{Significance}$

The main motivation of the paper is very important. Theoretical guarantees for solution methods with feasible at-scale practical implementations is highly desirable, since this informs us that they are generalizable to all problem instances, not just the particular settings in which they were tested in and shown good empirical results. Generalisation and theoretical guarantees in RL, particularly in policy-based methods with neural policy classes, which are the most successful algorithms used in practice, is a very important area of research.



**Weaknesses:**

$\textbf{Motivation}$

While the motivation of the paper is very important, i.e. insufficient theoretical guarantees for policy-gradient methods with neural function classes, PG have been shown to have global convergence beyond tabular fn classes which the authors claim to be the issue.  For log-linear policy classes see: Yuan  et al - “Linear Convergence of Natural Policy Gradient Methods with Log-Linear Policies” - and references within. For neural policy classes see “Neural Policy Gradient Methods: Global Optimality and Rates of Convergence” -- Wang et al and references within. Both of these show global convergence, the latter for a two-layer NN, including finite-sample guarantees/convergence rates and properties of actor-critic methods, with estimated critics, required for convergence.

$\textbf{Relation to RL and placement in context}$

The second main weakness of this paper is the lack of placement of their work in relation to the standard RL model used by the community through the formalism of MDPs or POMDPs, with accompanying class of solution methods. The paper would be significantly strengthened if they authors can clearly state what the limitations are with the previous class of problems described in RL via a reward signal and transition dynamics and in addition for POMDPs, how current definitions of high dimensional observation spaces proposed are better captured with their problem definition. It is unclear how the models proposed by this paper generalize beyond the log-linear policy class and action spaces of dimension 2, described within. The authors claim lack of theoretical work, yet previous work in RL has also analyzed distribution shift and generalization error, but with a different definition than the authors propose. It is unclear to me what the prior limitation in definition were and how this is a better way of capturing such quantities. Additional details in this respect would significantly strengthen the paper..

$\textbf{Purpose/Goal}$

Furthermore, it is rather confusing the purpose of the paper, i.e. it seems the authors compare problems against each other, for the same solution method, instead of comparing solution methods that are general enough to work on every problem instance. It is also confusing the setting and problem definition. It appears that the problem is not learning by trial-and-error, and that the problem instances proposed need heuristic descriptions of reward signals, horizon sizes and task termination. It is unclear the setting in which we are in, whether that is undiscounted finite-horizon, or continuing learning (infinite horizon, average reward). The authors reference terms from RL related to this but never actually formalize the problem.

$\textbf{Empirical study/experimental illustration}$

Lastly, the experimental section is performed on a certain game designed in particular way, which is an illustration rather than a practical algorithmic implementation, akin to the solution methods employed in empirical RL.

More details/questions in the next sections about these points.

I am happy to adjust my score if I have not correctly understood the paper, and the authors provide more details on how their work can be placed in the context of RL, which would help me understand the significance and impact of the result and analysis provided.

**Questions:**

$\textbf{Confusion w.r.t. setting}$

At times it is unclear which setting the authors analyze? Is it the finite-horizon setting? There is a significant emphasis on horizon size $T$, but this in their formulation is a hyperparmeter, which a solution method optimizes, not part of the problem definition and or the objective an algorithm is trying to optimize.

$\textbf{Confusion w.r.t. the feedback signal}$

Another confusion is the feedback signal.
The authors propose different heuristics for defining a feedback signal, but it is unclear what their connection is with the current reward signal in the standard MDP/POMDP model.
E.g. the number of times n, and the boolean task termination signal, the authors relate to sparsity, but it appears to also be related to the Markovian properties of the reward signal.
Isn’t a scalar reward signal enough even for continuing problems? Since the reward rate can adequately describe the class of lifelong learning problems in RL.

The authors mention the reward rate in the related work only, but throughout the paper they have an objective which they call “generalization error” or later redefined as “test error”. Next, the reward is given by a teacher represented as a neural network, where does this come from? The paper would be considerably strengthened by a clear comparison with the current model and a description of the limitations of the current model that they are trying to solve.

At line 109 - they say receiving penalties is not always beneficial, but this should be a problem definition, not available for an agent to change.
At line 167-which RL protocols? finite/infinite horizons? discounted/undiscounted(continuing), partially Markovian reward signals of certain orders?
Line 185 -”low asymptotic performance” — for a particular solution method?
Line 186 - “finite-size effects” - unclear what this means. Changing the reward changes the objective and the RL problem is known to be non-convex anyway. It is unclear how this analysis of the stationary points relates to what is known for policy-gradient methods.
Line 189 - “sub-reward” ?

$\textbf{Confusion on solution methods}$

The authors propose a particular model of policy gradients defined via a student network (or linear layer?) + a sigmoid activation function. It is unclear if their model/method extends to more general classes, particularly since in RL the class of solution methods extends to highly combinatorial action spaces or continuous action spaces, with log-linear, softmax, or neural function classes.

$\textbf{Confusing comparisons/hyperparameter optimization}$

 At times the authors compare different objective functions and say that a particular change in the reward or dynamics of the problem would yield better performance, which I find very confusing, since in general we are trying to compare the performance of different solution methods on the same problem/objective function, known or unknown. They then describe a tradeoff between problems, and how fast they are solved by the same solution method, or how well that solution method solves them (accuracy). They also appear to optimize over objectives, by doing sensitivity plots on the horizon size or the number of lives of an agent, and by comparing different feedback signals against each other claiming that one is better than another.
Line 198 - “optimal schedules for both hyper-parameters” - isn’t this problem dependent? Why is the episode length a hyperparameter? Why is this under the control of a solution method/algorithm/agent?

$\textbf{Confusing referrals to exploration}$

 At times the authors mention exploration, but I am very confused how a shallow single layer neural network with a sigmoid function is dealing with exploration. It is even unclear how exploration is defined for the optimization problem described by the “generalization error” objective function they propose. Generalization over what? Observations? Tasks? Environments?
Line 212 - “refined information” - wdym? Isn’t this changing the problem?

$\textbf{Confusion about phase transitions}$

Are these transitions between different objectives? Because they change based on the problem instance.
Line 253 “first-order phase transitions” - Not defined. I don't know what this means, I assume related to first-order stationary points of the objective function the solution method optimizes.

$\textbf{Confusion about speed-accuracy tradeoff}$

It is unclear the significance of this quantity. Doesn’t it actually tell you the speed with which different problems are solved by the same solution method that you chose? It is not a description of a solution method.

$\textbf{Clarity}$

At times the author use particular definitions without defining them, e.g., “reward stringency”, “order parameters”, “speed-accuracy” (speed of what? accuracy of what, w.r.t. what?).

$\textbf{Related work}$

The authors mention they are concerned with reward rate and episode length, but generally, the continuing setting using reward rates is the infinite horizon problem setting using average reward formulation (and thus reward rates) and has no episodes, by definition being continuing, never-ending, lifelong.
In the same section, the authors mention they focus on the average-case dynamics of policy-gradient methods, what does it mean “average-case”? There are reasons to use worst-case guarantees, they tell us the behaviour of an algorithm on all problem instances. Is the average case, an expectation over problems?

The authors make certain referrals to terms from other fields, like statistical mechanics or heuristics from statistical physics which may be unknown to the reader and unclear on their significance. If they are important, they should be defined.

$\textbf{Experimental section}$

Isn’t the dimensionality of the observation space very small to be representative of “high dimensional” observation spaces?
Line 292: what is a pure random policy? Does this mean this is off-policy learning from a behaviour policy that is random? Wouldn’t that mean that the gradient-update is biased?
The problem instance seems very particular. It is unclear how this particular problem instance reflects the generality of the claims set in the introductory contributions.

$\textbf{Miscellaneous}$

"D" is not defined, though from context it appears to be the dimensionality of the observation space.
Typo at line 250. Sentence doesn’t make any sense. “Learning with $\eta_2 This is not a …”
Typo at line 246 “the the”


**Limitations:**

$\textbf{Lack of clear description of the limitations}$

The authors do not provide a clear description of the limitation of their model. It is unclear what exactly is “solvable” in closed form, is it any problem of any dimensionality with any kind of reward signal, including non Markovian? How general is the problem that is “solvable”? Does it capture all problems a POMDP would capture, and these problems are solvable? Any kind of additional information in this respect would be useful for the reader. It is also unclear what is “the average-case”. A short definition would be very useful.

---

> ### Author Rebuttal · Authors · 2023-08-09
>
> We thank the reviewer for their time spent reading our paper, and for the thoroughness of the review. Due to character limits, we only briefly respond to some points, but we are happy to elaborate during the discussion period.
>
> Clarity is paramount and we clearly can improve thanks to the feedback. We believe some of the reviewer’s comments stem from the impression that the paper offers a new formalism or solution method applicable in practice; this is not its primary purpose.Instead, its goal is to provide insight into an existing algorithm by exhaustively studying a simple (and hence mathematically tractable) case, an approach that has borne fruit in supervised learning, see the reviews [1,2] for an overview of classic results, and [3-7] for a small subset of recent NeurIPS/ICML/ICLR papers in this direction. Because this approach is new in RL, we will include a more extensive intro and discussion of the points below in the revision. Please let us know if you have any follow-up questions:
>
> * **Our goal** is to develop a theory for the typical dynamics of the REINFORCE policy gradient algorithm. For example, we would like to explain how properties of a problem and algorithmic choices impact how quickly a model will learn, or how effectively it will generalise.
>
>     **Our approach** is to consider a simplified model of a reinforcement learning problem. These simplifications allow exact solutions, i.e. we derive a set of equations that describe the typical learning dynamics exactly, and finally we analyse the properties of the solution.
>
>     We **do not propose a new method to tackle RL problems; we also do not investigate worst-case performance guarantees**; instead, we mathematically derive the typical properties of learning with policy gradient in a simple case, in order to better understand it.
>
>
> * **Relation to works showing global convergence**: Thank you, we will add these citations. Our contribution differs by exactly characterising the typical learning dynamics at all times, rather than at the end of training (convergence).
>
>
> * **Relation to MDP / POMDP modelling**: in Appendix B we describe our setting as a standard finite-horizon POMDP. We will incorporate this discussion into the main text.
>
> * **The role of experiments**: As with all theoretical studies at present, we inevitably make simplifying assumptions. Our experiments verify whether the qualitative insights gained from our analysis carry over to more realistic settings. Indeed, as shown in fig. 10 of the Appendix, we find the speed-accuracy trade-off predicted by our theory in Pong (also shown in fig. D in the attachment) using the REINFORCE algorithm in a deep network trained with ADAM. We will move this figure to the main text. During the rebuttal period we have also obtained new results with REINFORCE on Bossfight (Fig. A of the attachment).
> # Questions
> * **Setting**: Undiscounted finite horizon with episode length $T$.
> * **Feedback signal**: Our approach examines how different feedback signals will affect the performance of a learner. As is common in theory of this type, the rewards of the network that is trained (called the *student* following the theory for supervised learning [1-7]) are determined by another neural network, called the *teacher*, which is a theoretical stand-in for the target policy. This teacher provides rewards for newly sampled inputs, enabling calculation of generalisation behaviour (more below).
>     * **Penalties are not always beneficial/protocols/sub-reward**: Yes, our goal is to trace out learning behaviour for diverse reward schemes, not all beneficial. Protocols/subrewards correspond to possible reward functions.
>     * **Performance**: generalisation performance.
>     * **Finite-size effects**: deviations between simulations with a fixed input dimension and our theory, derived in the limit where the input dimension goes to infinity.
>
> * **Generalisation error**: Expected performance of a policy on a new episode. Because observations are sampled, every observation is almost guaranteed to be unique (generalisation is over observations).
>
> * **Average-case**: The average behaviour one would expect from many learning instances of the same problem, rather than the worst case.
>
> * **Solution methods**: We consider a simple log-linear policy network, analogously to how studies of this type began with the perceptron in supervised learning. We expect that future work can generalise to multiclass and deeper networks, building on these methods.
>
> * **Comparisons/Hyperparameter optimization**:  We compare the performance of a single solution method on different objective functions/rewards/algorithm hyperparameters, to systematically understand learning behaviour.
>
> * **Exploration** arises from random sampling of observations.
>
> * **Phase transition**: qualitatively different behaviours in regions of problem instances (here the (reward,penalty) plane). We will define the term ‘first-order-phase transition.’
>
> * **Speed-accuracy tradeoff**: ‘Doesn’t it actually tell you the speed [and final accuracy] with which different problems are solved by the same solution method?’ - Yes exactly.
>
>
> * **Experimental section**: The dimension (6720) is representative of high-dimensional observation spaces.
>
>
> * **Reward stringency**: the strictness of requirement for a reward, e.g. all decisions correct is more stringent than n or more; **order parameters**: variables defined in eq. 3
>
>
> * **Miscellaneous**: $D$ is input dimension. Thanks for the typos.
>
>
> **Limitations**: We will collect our discussion of limitations in a paragraph of the conclusion, listing: our focus on binary action spaces, the environment data being sampled from Gaussian distributions, the focus on shallow networks, and that we do not consider state transitions conditional on action, which are all things we are looking to address.

---

> > ### Comment · Reviewer_n4bo · 2023-08-14
> > **Rebuttal Response**
> >
> > Thank you for the clarifications and the rebuttal response.
> >
> > After reading the authors’ response I believe my initial understanding of the paper was correct in that the authors are showing properties of the problem (if the reward function or the episode length change) coupled with a particular solution method (author’s say it is the REINFORCE algorithm). I really hope the authors clarify this very important information in the paper as none of these was clear in the original manuscript (motivation, scope, purpose, problem formulation, the setting in which they operate, which I know understand to be the <<undiscounted finite horizon with episode length T>>)
> >
> > I never believed this paper to be providing a new solution, but I also do not completely understand the insight it would bring to RL algorithms per se. REINFORCE is not really a practical algorithm. Among the reasons is the finite horizon setting of a certain length, and sample inefficiency due to that, particularly since in practice these values need to be very large and are ill-suited for this algorithm when paired with sparse rewards.
> > Afaik it also globally converges sublinearly for all problem instances. Log-linear policies afaik are relatively well understood and proven globally convergent with known rates, even for actor-critic algorithms with value estimated critics.
> >
> > I also do not fully understand what the average case really means (the explanation provided is not rigorous) if we are talking about problems not solution methods. Since we do not have any control over problems, how can there be an average case?
> >
> > Our contribution differs by exactly characterising the typical learning dynamics at all times, rather than at the end of training (convergence).
> > Isn’t this what sample complexity results also show? Asymptotic complexity is something else, but iteration/sample complexity would show how learning behaves at any time.
> >
> > Because observations are sampled, every observation is almost guaranteed to be unique (generalisation is over observations).
> > So observations are in fact states? Are you investigating MDPs?
> >
> > $\textbf{Updating my score}$
> > I have read all other reviews and the rebuttals.
> > It is not clear to me that this approach will provide more insight into RL algorithms or MDP problems when used in RL algorithms, nor do I see how this explains the behavior of commonly employed methods used in practice.
> > However, I see that the work has value to the community since other reviewers have found it useful and interesting, so I will increase my original overall score.

---

> > > ### Author Response · Authors · 2023-08-18
> > >
> > > We thank you for your continued engagement with our paper, and for updating your score. We do indeed plan on revising the manuscript to improve its clarity following the suggestions from the reviews. As for your questions / points (which we split into two posts due to character restrictions):
> > >
> > > * **REINFORCE is not really a practical algorithm**: Our overarching objective is to advance our understanding of learning dynamics in deep reinforcement learning (RL) systems. While we acknowledge that REINFORCE is a simple algorithm, we view its analysis as an essential first step toward achieving this overarching goal because REINFORCE is the foundation of all other policy-gradient algorithms. Meanwhile, for supervised learning a huge effort is still ongoing to analyse the dynamics of vanilla SGD for a single perceptron -- see for example the paper by Ben Arous et al. [11] that won “Outstanding paper” at NeurIPS 2022 (no momentum, no adaptive learning rates as in Adam, etc.) For REINFORCE, none of this type of analysis had been carried out yet.
> > >
> > > * **Average-case**: Learning is a stochastic process -- if you run an experiment several times using the same learning-rule, rewards given and hyperparameters, there will be differences due to the different episodes experienced by the learner, etc. Yet some key quantities, such as the performance of the learner, will evolve in a predictable way, following closely the average behaviour one would expect from many learning instances of the same problem. Large deviations from this behaviour occur only very rarely. The strength of our method is then that we make statements about this average case, which tend to be more precise than worst-case type of bounds, which have to be based on the worst deviation, no matter how unlikely. This is the average case we’re referring to. On a technical level, the ODE description of the learning dynamics hinges on the concentration of the order parameters - i.e how quickly the sums in eq 3 converge to their mean -  which makes it possible to map the stochastic evolution of the student onto a deterministic evolution of the order parameters.
> > >
> > >     **Why is average-case important in practice?**: Analysing the average case in our model makes concrete practical predictions, for example the speed-accuracy trade-off, which could not be predicted using methods that analyse the worst case. The validation of these predictions in experiments with procedurally generated problems (bossfight and PONG) highlights the usefulness of this type of analysis in practice.
> > >
> > >     Analysis of the ‘average-case’ has a long history in theoretical machine learning and has been a focus for the statistical physics of learning community for supervised learning problems. This analysis is of benefit/complementary to complexity bounds provided by statistical learning theory, which tend to be overly-pessimistic and not characterise the average or most probable behaviour.
> > >
> > > [11] Gerard Ben Arous, Reza Gheissari, & Aukosh Jagannath (2022). High-dimensional limit theorems for SGD: Effective dynamics and critical scaling. In Advances in Neural Information Processing Systems.
> > >
> > > 1/2

---

> > > > ### Author Response · Authors · 2023-08-18
> > > >
> > > > 2/2
> > > >
> > > > * **Connection with MDPs**: MDP and POMDP provide mathematical models of decision making in problems that can be addressed using reinforcement learning. Here instead, we are interested in analysing the simplest RL algorithm, REINFORCE, on a given problem. The problem is designed to both (1) allow for a precise description of the learning dynamics of the neural network trained on this problem (sec. 2.1) while (2) maintaining the phenomenology of deep neural networks trained on more complex problems, which we verify with our experiments in sec. 3. We agree, however, that it could have been beneficial to explicitly show how our framework can be incorporated within the standard formalism of POMDPs. This is addressed in Appendix B and illustrated in fig 1b, where the notion of states, state transitions and observations are addressed. ‘So observations are in fact states? Are you investigating MDPs?’: It is true, an alternative interpretation is possible in which one considers the observations as states. Although, in this case, the MDP and POMDP formulations are different ways of describing the same thing, they both lead to different useful extensions: with the MDP version the featurisation can be altered, and an interesting line of work would be to extend to nonlinearly-separable featurizations and deeper networks. Whereas the POMDP version would be a much more natural one to extend to more (than 2) low-dimensional latent states and consider action-dependent transitions between them, which is an extension we are actively looking at.
> > > >
> > > > * **Log-linear policies**: We do not see our method as being restricted to model log-linear policies: The policy is defined by a student network, consisting of a perceptron (linear layer+sign non-linearity). The perceptron (the basic building block of a neural network) is used as a simple neural function class amenable to exact analysis. We show equivalence to a policy parameterised by the same network in the early stages of training. This is a first step in being able to model more complex neural function classes, with more layers, and is a method that has borne fruition in the supervised setting - we start with the perceptron, and gradually add complexity. Although the *convergence* properties of log-linear policies are fairly well understood, we again are concerned with the entire learning trajectory.
> > > >
> > > >
> > > > * **Behaviour of commonly employed methods used in practice**: As argued in the reply to the other reviewers, it is possible to extend our model to problems with more states and more decisions using results from statistical physics [10]. From there it would be possible to define the notion of ‘value’ on states/actions, meaning there is the potential to incorporate value-based RL algorithms or algorithms that combine policy and value-based methods (like the actor-critic), and this is a plan for future works. It would also be possible to extend to higher-dimensional action spaces instead of binary by considering work that finds learning curves for the multi-class perceptron [8]. This again would widen the possibilities of RL agents we can consider, and also widen the number of suitable environments we can test against. We will discuss these avenues for further work in the revised manuscript.

---

> > > > ### Comment · Reviewer_n4bo · 2023-08-19
> > > > **average-case and variance**
> > > >
> > > > Thank you for the clarifications.
> > > > Should I understand that the average-case ignores the variance over problem instances?
> > > > And the analysis shows how the system evolves over time, but only the average of the distribution over problems?

---

> > > > > ### Author Response · Authors · 2023-08-20
> > > > >
> > > > > Thank you for your continual engagement. We would like to answer in two points. But first must clarify what you mean by ‘variance over problem instances’. The average-case analysis is done with respect to a **single** problem, i.e. the average behaviour one would expect if they were to run infinite instances of the **same** problem (same hyperparameters, episode length, learning-rule and rewards).
> > > > >
> > > > > * In theoretical analysis, the average-case does **not ignore the variance** in behaviour for a problem - in fact the order parameters (eq. 3), by definition, are the variance and covariance of the neural pre-activations of the teacher and student (see section 2.1). The order parameters (and also the performance) converge to their mean value with a variance that decays as $\mathcal{O}(1/D)$, so naturally we describe how the mean performance evolves.
> > > > >
> > > > > * In our practical experiments with Bossfight and PONG, there is of course some finite variance, which we indicate with the shaded areas in Figs A and D of the attachment; however, we find that the behaviour of the mean performance over several runs matches some of the predictions of our theoretical model qualitatively
> > > > >
> > > > >
> > > > > * **‘And the analysis shows how the system evolves over time, but only the average of the distribution over problems?’:** The analysis does indeed show how the average behaviour of the system evolves over time. But the average is over the distribution of the **same** problem. And the system converges to its average behaviour with variance that decays as $\mathcal{O}(1/D)$ (in the theoretical analysis).

---

> > > > > > ### Comment · Reviewer_n4bo · 2023-08-21
> > > > > >
> > > > > > Thank you for the clarifications.
> > > > > > I was confused because you said it somehow offers more than the standard analysis, which can also be in expectation over any stochasticity from randomly sampled trajectories, and can also include bias and variance analysis from such sampling.

---

> > > > > > > ### Author Response · Authors · 2023-08-21
> > > > > > >
> > > > > > > We again thank the reviewer for their continual engagement, it's greatly appreciated.
> > > > > > >
> > > > > > > To the best of our knowledge, other methods that compute expected behaviour over the stochastic process described by the learning dynamics are limited either to low-dimensional problems, or limited in the observables that they can characterise (e.g., they can't access the expected reward at all times), or limited in the phase of learning that they can access (e.g. only the large time convergence behaviour). Our method doesn't suffer from such limitations (we are able to characterize the dynamics of relevant observables at **all** times in **high** dimensions). In case the reviewer knows other results that have such generality, we kindly ask to point them out. They would represent an important addition to our "further related works" section, and it is important to contrast our results to theirs.

---

> > > > > > > > ### Comment · Reviewer_n4bo · 2023-08-22
> > > > > > > >
> > > > > > > > What I was referring to are results like the one below, which shows sample complexity bounds for general policy-based methods with linear function approximation, which bound the bias/variance/number of samples/rate of convergence in the policy based on the same characterisation of convergence of the critic. The results are finite sample bounds, not asymptotic results, and closely follow algorithmic implementations, based on off-policy sampling, and value-based algorithms.
> > > > > > > >
> > > > > > > > @misc{chen2023approximate,
> > > > > > > >       title={An Approximate Policy Iteration Viewpoint of Actor-Critic Algorithms},
> > > > > > > >       author={Zaiwei Chen and Siva Theja Maguluri},
> > > > > > > >       year={2023},
> > > > > > > >       eprint={2208.03247},
> > > > > > > >       archivePrefix={arXiv},
> > > > > > > >       primaryClass={cs.LG}
> > > > > > > > }
> > > > > > > >
> > > > > > > > I don't exactly understand what "at all times means" or the techniques you use so I cannot compare.

---

### Author Rebuttal · Authors · 2023-08-09

We thank the reviewers for their time spent and comments on our paper. Here is a list of references referred to in other responses, and attached is the PDF of new figures.

[1] Andreas Engel and Christian Van den Broeck. Statistical mechanics of learning Cambridge University Press, 2001

[2] Marylou Gabrié, Surya Ganguli, Carlo Lucibello, and Riccardo Zecchina. Neural networks: from the perceptron to deep nets. ArXiv preprint, abs/2304.06636, 2023. URL https://arxiv.org/abs/2304.06636

[3] Ben Sorscher, Surya Ganguli, and Haim Sompolinsky. Neural representational geometry underlies few-shot concept learning. Proceedings of the National Academy of Sciences, 119(43): e2200800119, 2022. doi: 10.1073/pnas.2200800119. URL https://www.pnas.org/doi/abs/ 10.1073/pnas.2200800119

[4] Lampinen and Ganguli. An analytic theory of generalization dynamics and transfer learning in deep linear networks. ICLR 2019 https://openreview.net/forum?id=ryfMLoCqtQ.

[5] Lee et al. Continual learning in the teacher-student setup: Impact of task similarity. ICML 2021 http://proceedings.mlr.press/v139/lee21e.html


[6] Ben Arous, Gheissari, Jagannath: High-dimensional limit theorems for SGD: Effective dynamics and critical scaling. https://proceedings.neurips.cc/paper_files/paper/2022/hash/a224ff18cc99a71751aa2b79118604da-Abstract-Conference.html

[7] Veiga et al. Phase diagram of Stochastic Gradient Descent in high-dimensional two-layer neural networks. NeurIPS ‘22. https://proceedings.neurips.cc/paper_files/paper/2022/hash/939bb847ebfd14c6e4d3b5705e562054-Abstract-Conference.html

[8] Cornacchia et al. Learning curves for the multi-class teacher–student perceptron. Mach. Learn. Sci. Technol. 4 (2023) 015019 https://iopscience.iop.org/article/10.1088/2632-2153/acb428/pdf

[9] Chiyuan Zhang, Samy Bengio, Moritz Hardt, Benjamin Recht, and Oriol Vinyals. 2021. Understanding deep learning (still) requires rethinking generalization. Commun. ACM 64, 3 (March 2021), 107–115. https://doi.org/10.1145/3446776

[10] Loureiro, Bruno, et al. "Learning gaussian mixtures with generalized linear models: Precise asymptotics in high-dimensions." Advances in Neural Information Processing Systems 34 (2021): 10144-10157.

---

### Decision · Program_Chairs · 2023-09-21

**Decision:**

Reject

**Comment:**

There was some disagreement among reviewers in terms of clarity and experiments.
- There is confusion among reviewers for the clarity of the work and its value to the community (n4bo, vS6U, vS6U, 9GSU). Some reviewers suggested a significant rewrite of the work and to include many more experiments. Last, a collection of these reviewers are unclear on the paper's goals.
- While some reviewers have noted their approval for the paper they have also listed low confidence in their scores (bMDj, nMd7)

 After further review of the paper, the paper does appear difficult to follow. It should be reorganized to be precise to the quality of a NeurIPS paper.

Here is a list of proposed updates that are important changes to ensure the community fully appreciates the work.
1. Figure 1 appears far too early in the paper. Much of the concepts, notation, and equations referenced inside of the caption are not explained until later, making it extremely challenging to be able to understand the overall method displayed.
2. It's also unclear why the RL perception framework seems to be described in the introduction and section 2. This is likely to confuse readers of the paper.
3. In the introduction, it's described that prior methods have used statistical tools for analysis. Still, the paper does not say how these statistical tools will help RL researchers or motivate the application of this perceptron-type analysis method and how it can be helpful, especially in the high dimensional case for reinforcement learning applications. Increasing the clarity around these issues will make it easier to see the value of this work for the community.


Technical Clarity:
1. The paper states that it's crucially important that the student does not have access to the correct action. In this context, in section one it's not clear why this is so important for the method.
2. To help connect the proposed method to prior reinforcement learning literature, the Indicator function and Criterion in equation 1 could be compared to an advantage function that is used often in policy grading update methods. The direct comparison is helpful, but REINFORCE is an old algorithm. The work lacks connections to more recent RL algorithms, such as PPO. Reviewer 9GSU notes something similar.
3. The terms Q and R that are used in the method formulation will likely confuse the reinforcement learning community. Often, these terms are used to describe the Q function and the return values R. To increase the likelihood that reviewers from the reinforcement learning community understand the overall work, it would be best to choose other symbols. Reviewer n4bo notes something similar.
4. The author should describe in more detail why the three different learning protocols correspond to their names used in the paper. How does the setting for the specific type of ODE result in the notion of breadcrumbs?
5. Explain how the student-teacher overlap is strongly related to generalization. Also, provide details on the type of generalization.

Experiments:
Also, the experiments section is limited to the point where the claims may only be true for a subset of applications. For example, there do not appear to be any details in the first half of the paper about the type of reinforcement learning task that is being used for evaluation in these experiments. It is necessary to describe which task is being used for evaluation to understand the scope of the impact of the proposed method. This is done for the experiments section but this section only includes an analysis of a single task, BossFight. Many more details are required to understand the experiment section. For one, there is no specific algorithm described in the paper for how to use this method to form the type of analysis proposed. For example, the details of the neural network are not described. How are actions generated from a "pure random policy"? Why are the agent pixels set to zero? It is not clear what this pixel zeroing means.

Lastly, the proposed method may have connections to illustrating some of the similarities in the dynamics of the training problem that is known for more empirical analysis, however, the proposed method does not suggest how to use this analysis to improve current reinforcement learning algorithms. This would be an important contribution of the method that would increase its value and should be compared to recent work on studying RL learning dynamics and Generalization [1].

The paper can increase its future impact by showing the relationship between the proposed model and common deep RL methods.

References:
1. Learning Dynamics and Generalization in Deep Reinforcement Learning. Clare Lyle, Mark Rowland, Will Dabney, Marta Kwiatkowska, Yarin Gal Proceedings of the 39th International Conference on Machine Learning, PMLR 162:14560-14581, 2022.